

# A new method for atmospheric detection of the CH₃O₂ radical

Lavinia Onel[1], Alexander Brennan[1], Paul W. Seakins[1,2], Lisa Whalley[1,2], Dwayne E. Heard[1,2]

[1]School of Chemistry, University of Leeds, Leeds, LS2 9JT, UK
[2]National Centre for Atmospheric Science, University of Leeds, Leeds, LS2 9JT, UK

*Correspondence to*: Lavinia Onel (chmlo@leeds.ac.uk); Dwayne Heard (d.e.heard@leeds.ac.uk)

**Abstract.** A new method for measurement of the methyl peroxy ($CH_3O_2$) radical has been developed using the conversion of $CH_3O_2$ into $CH_3O$ by excess NO with subsequent detection of $CH_3O$ by fluorescence assay by gas expansion (FAGE) with laser excitation at *ca.* 298 nm. The method can also directly detect $CH_3O$, when no nitric oxide is added. Laboratory calibrations were performed to characterise the FAGE instrument sensitivity using the conventional radical source employed in OH

calibration with conversion of a known concentration of OH into $CH_3O_2$ via reaction with $CH_4/O_2$. Detection limits of $3.8 \times 10^8$ molecule cm⁻³ and $3.0 \times 10^8$ molecule cm⁻³ were determined for $CH_3O_2$ and $CH_3O$, respectively for a signal-to-noise ratio of 2 and 5 min averaging time. Averaging over 1 hour reduces the detection limit for $CH_3O_2$ to $1.1 \times 10^8$ molecule cm⁻³ comparable to atmospheric concentrations. The kinetics of the second–order decay of $CH_3O_2$ via its self–reaction were observed in HIRAC (Highly Instrumented Reactor for Atmospheric Chemistry) at 295 K and 1 bar and used as an alternative

method of calibration to obtain a calibration constant with overlapping error limits at the $1\sigma$ level with the result of the conventional method of calibration. The overall uncertainties of the two methods of calibrations are similar: 15 % for the kinetic method and 17 % for the conventional method and are discussed in detail. The capability to quantitatively measure $CH_3O$ in chamber experiments is demonstrated via observation in HIRAC of $CH_3O$ formed as a product of the $CH_3O_2$ self–reaction.

**1 Introduction**

Methyl peroxy ($CH_3O_2$) radicals are critical intermediates in the atmospheric oxidation (Orlando and Tyndall, 2012) and combustion of hydrocarbons (Zador et al., 2011). In the remote atmosphere $CH_3O_2$ is mainly formed by the reaction of methane with the OH radical via abstraction of an H atom (R1), followed by the reaction of the produced $CH_3$ radical with $O_2$ (R2).

$OH + CH_4 \rightarrow CH_3 + H_2O$     (R1)

$CH_3 + O_2 + M \rightarrow CH_3O_2 + M$     (R2)

Methyl radicals can also be formed from more complex species, e.g. the reaction of acetyl peroxy radicals with $HO_2$ in low $NO_x$ environments or the reaction of acetyl peroxy radicals with NO in anthropogenically influenced environments. $CH_3O_2$ is

predicted to be the most abundant peroxy radical in the atmosphere, yet there are no specific measurements of its concentration. Daytime concentrations estimated using a box model utilizing the MCM (Master Chemical Mechanism) version 3.3.1 (Saunders et al., 2003;Jenkin et al., 2015) are ~ $6 \times 10^8$ molecule cm⁻³ in the tropical Atlantic ocean in summer (Whalley et al., 2010), ~ $2 \times 10^8$ molecule cm⁻³ in a tropical rainforest (Whalley et al., 2011), and lower in polluted environments, for example ~ $5 \times 10^7$ molecule cm⁻³ in London in summertime (Whalley et al., to be submitted).

The reaction of $CH_3O_2$ with NO (R3) usually dominates the chemistry of $CH_3O_2$, particularly in environments influenced by anthropogenic $NO_x$ emissions, resulting in $NO_2$ production and hence ozone production:

$CH_3O_2 + NO \rightarrow CH_3O + NO_2$     (R3)



The subsequent reaction of $CH_3O$ with $O_2$ (R4) produces $HO_2$, which in turn oxidises another NO to $NO_2$ (R5) with further production of $O_3$ and propagation of the $HO_x$ radical chain:

$$CH_3O + O_2 \rightarrow CH_2O + HO_2 \qquad\qquad (R4)$$

$$HO_2 + NO \rightarrow OH + NO_2 \qquad\qquad (R5)$$

However, under low $NO_x$ levels (e.g. remote forested environments and the marine boundary layer) the self-reaction of $CH_3O_2$ (R6) and the reactions of $CH_3O_2$ with $HO_2$ and other organic peroxy ($RO_2$) species are important radical removal/termination reactions. The $CH_3O_2$ self-reaction occurs through two channels, (R6.a) and (R6.b) (Tyndall et al., 1998):

$$CH_3O_2 + CH_3O_2 \rightarrow CH_3OH + CH_2O + O_2 \qquad\qquad (R6.a)$$
$$CH_3O_2 + CH_3O_2 \rightarrow CH_3O + CH_3O + O_2 \qquad\qquad (R6.b)$$

Despite the importance of the reaction (R6), there are uncertainties of about a factor of two in the value of its rate coefficient

at room temperature, $k_6$, which ranges from $(2.7-5.2) \times 10^{-13}$ cm$^3$ molecule$^{-1}$ s$^{-1}$ (Atkinson et al., 2006); the preferred IUPAC value is $k_6 = 3.5 \times 10^{-13}$ cm$^3$ molecule$^{-1}$ s$^{-1}$ (Atkinson et al., 2006). The previous kinetic studies used time-resolved UV-absorption spectroscopy to detect $CH_3O_2$ radical, typically at 250 nm, (Sander and Watson, 1980, 1981;McAdam et al., 1987;Kurylo and Wallington, 1987;Jenkin et al., 1988;Simon et al., 1990;Lightfoot et al., 1990). UV-absorption spectroscopy is a relatively insensitive technique and hence the detection limits of $CH_3O_2$ were quite high, for example approximately $4 \times$

$10^{12}$ molecule cm$^{-3}$ (Sander and Watson, 1980, 1981). In addition, due to the broad, featureless spectra of $RO_2$ species, which often overlap, UV-absorption is a relatively unselective technique for the study of the kinetics of individual $RO_2$. Therefore, there is a clear need for the determination of $k_6$ using a more selective method, which will be addressed in subsequent studies.

At present, $CH_3O_2$ is not specifically measured in the atmosphere by any direct or indirect method. Time-resolved continuous-wave cavity ringdown spectroscopy (CRDS), using the $\nu_{12}$ transition of the $A \leftarrow X$ band at $\sim 1.3$ μm has been used

to detect $CH_3O_2$ directly in a photoreactor (Farago et al., 2013;Bossolasco et al., 2014). However, the detection limit is not sufficiently sensitive to enable tropospheric detection. Typically, the sum of $HO_2$ and all organic $RO_2$ has been measured in the atmosphere, making no distinction between $HO_2$ and different $RO_2$ species, although more recently the sum of $RO_2$ has been quantified separately to $HO_2$. One of the methods uses Chemical Ionisation Mass Spectrometry to determine the sum $[HO_2] + \sum_i[RO_{2,i}]$ or separately $[HO_2]$, depending on the control of the flows of the NO and $SO_2$ reagents (Hanke et al.,

2002;Edwards et al., 2003). The sum $[HO_2] + \sum_i[RO_{2,i}]$ has also been determined for many years by the Peroxy Radical Chemical Amplifier (PERCA) method, which uses NO and CO to generate $NO_2$ amplified by a chain reaction, and subsequently measured by a variety of methods, for example luminol fluorescence, laser-induced fluorescence (LIF) or cavity absorption methods (Cantrell and Stedman, 1982;Cantrell et al., 1984;Miyazaki et al., 2010;Hernandez et al., 2001;Green et al., 2006;Chen et al., 2016). A modification of PERCA, using a denuder to remove $HO_2$ has been used to estimate the sum of

$RO_2$ (Miyazaki et al., 2010). $RO_xLIF$ is a more recent method, which uses OH LIF detection at low pressure, known as FAGE (fluorescence assay by gas expansion) (Fuchs et al., 2008;Whalley et al., 2013). The $RO_xLIF$ method measures either $[HO_x] = [OH] + [HO_2]$ by converting $HO_x$ into $HO_2$ through addition of CO, or $[RO_x] = [HO_x] + \sum_i([RO_{2,i}] + [RO_i])$ by titrating $RO_x$ to $HO_2$ by added NO and CO. After the conversion into $HO_2$, $HO_2$ is converted into OH in the FAGE chamber and detected by LIF. The sum $\sum_i[RO_{2,i}]$ and the concentration of the initial $HO_2$ can be determined from the separate measurements of $HO_x$,

$RO_x$ and OH. The limit of detection of the $RO_xLIF$ method is $\sim 0.1$ pptv ($2.5\times10^6$ molecule cm$^{-3}$) (Fuchs et al., 2008;Whalley et al., 2013). Recently, the interference from certain types of $RO_2$ radicals in the FAGE detection of $HO_2$ was deliberately exploited to enable a partial $RO_2$ speciation (Whalley et al., 2013). The method was used in the Clean Air for London campaign



(ClearfLo) to distinguish between the sum of alkene, aromatic and long-chain alkane-derived $RO_2$ radicals and the sum of short-chain alkane-derived $RO_2$ radicals (Whalley et al., 2013).

As methoxy ($CH_3O$) radicals can be generated by techniques such as pulsed laser photolysis and microwave discharge and detected with high sensitivity by LIF (Shannon et al., 2013;Chai et al., 2014;Albaladejo et al., 2002;Biggs et al., 1993;Biggs

et al., 1997), the method has been used in kinetic studies of a range of $CH_3O$ reactions. These studies used the electronic excitation of the methoxy radical from the ground state to the first electronically excited state ($A^2A_1 \leftarrow X^2E$). The $A \leftarrow X$ excitation spectrum covers the range ~ 275–317 nm and leads to fluorescence from several vibronic bands in the near UV, and has been reported in a series of experimental and theoretical studies (Inoue et al., 1980;Kappert and Temps, 1989;Powers et al., 1997;Nagesh et al., 2014).

This paper reports the development of a new method for the selective and sensitive detection of $CH_3O_2$ radicals using FAGE by titrating $CH_3O_2$ to $CH_3O$ by reaction with added NO (R3) and then detecting the resultant $CH_3O$ by off-resonant LIF with laser excitation at *ca*. 298 nm. The method is similar to the standard method used for the detection of $HO_2$ radicals by FAGE through conversion of $HO_2$ to OH by reaction with added NO followed by OH on-resonance LIF at about 308 nm (Heard and Pilling, 2003). As LIF is not an absolute detection method, FAGE instruments require calibration, with the 184.9

nm photolysis of water vapour in air using a mercury (Hg) Pen-Ray lamp being a common method employed for generating known concentrations of OH and $HO_2$ (Heard and Pilling, 2003):

$$H_2O \xrightarrow{184.9 \text{ nm}} OH + H \tag{R7}$$
$$H + O_2 + M \rightarrow HO_2 + M, \tag{R8}$$

where M = $N_2$, $O_2$ and the photodissociation quantum yield of OH and H is unity. In this study the photolysis of water vapour is performed in the presence of excess methane to produce $CH_3O_2$:

$$CH_4 + OH \rightarrow CH_3 + H_2O \tag{R1}$$
$$CH_3 + O_2 + M \rightarrow CH_3O_2 + M \tag{R2}$$

An alternative $CH_3O_2$ calibration is also presented, consisting of the analysis of the kinetics of the $CH_3O_2$ decay by self-reaction monitored by FAGE and compared with the water photolysis method. The studies are performed within HIRAC (Highly Instrumented Reactor for Atmospheric Chemistry) which is a 2.25 $m^3$, custom-built, stainless steel chamber simulating

the ambient conditions (Glowacki et al., 2007). HIRAC has been used in alternative calibrations of FAGE for OH and $HO_2$ using the temporal evolution of appropriate species, in validation and development of new atmospheric measurement techniques as well as in kinetic and mechanistic studies of atmospheric relevant reactions (Malkin et al., 2010;Winiberg et al., 2015;Winiberg et al., 2016).

Direct LIF detection of $CH_3O$ radicals, which is also a key intermediate in the oxidation of methane and other VOCs in the

troposphere and formed by reactions such as (R3) and (R6.b), is also reported here. However, in the atmosphere $CH_3O$ is exclusively consumed by reaction with $O_2$ (R4) generating formaldehyde and recycling $HO_2$, resulting in a very short lifetime and consequently very low concentration (~$10^2$-$10^3$ molecule $cm^{-3}$). For this reason no measurements in the atmosphere have previously been attempted. The photolysis of $CH_3OH$ at 184.9 nm is used to estimate the FAGE sensitivity for $CH_3O$. The dominant photolysis channel of methanol between 165 and 200 nm generates $CH_3O$ radicals (Wen et al., 1994;Kassab et al.,

1983;Marston et al., 1993):

$$CH_3OH \xrightarrow{165-200 \text{ nm}} CH_3O + H \tag{R9}$$





A photodissociation quantum yield of $CH_3O$ of $0.86 \pm 0.10$ has been found at 193.3 nm (Satyapal et al., 1989) in qualitative agreement with analysis of the end-products of the methanol photodissociation at 184.9 nm (Porter and Noyes, 1959;Buenker et al., 1984). Here we report he first measurements of $CH_3O$ concentrations in an atmospheric simulation chamber. Methoxy

radicals are generated by the $CH_3O_2$ self–reaction carried out within HIRAC at 295 K and 1000 mbar of $N_2$ containing $O_2$ in trace amounts to reduce the rate of removal of $CH_3O$ by reaction with $O_2$. This work enhances the capability of HIRAC to measure short–lived radical species by the addition of both $CH_3O_2$ and $CH_3O$ detection, and we discuss the potential of the method for detection of $CH_3O_2$ in the atmosphere itself.

## 2 Experimental

**2.1 The FAGE instrument**

Details on the HIRAC-based FAGE instrument for the detection of OH and $HO_2$ has been presented previously (Winiberg et al., 2015). Figure 1 shows a schematic cross-section of the instrument inlet and the two fluorescence detection cells. The gas was sampled through a 1 mm diameter pinhole and passed down a 50 mm diameter flow tube of 280 mm length first into the OH detection axis and, after a further 300 mm, into the $CH_3O_2$ detection axis. The pressure in the detection cells was maintained

at $(2.65 \pm 0.05)$ Torr by using a high capacity rotary-backed roots blower pumping system (Leybold, trivac D40B and RuVac WAU251). $CH_3O_2$ radicals were titrated to $CH_3O$ by adding high purity NO (BOC, N2.5 Nitric Oxide) with a typical 2.5 sccm flow rate (further details in Section 2.2) ~25 mm before the second detection axis into the centre of the flow. The resultant $CH_3O$ radicals were measured by LIF.

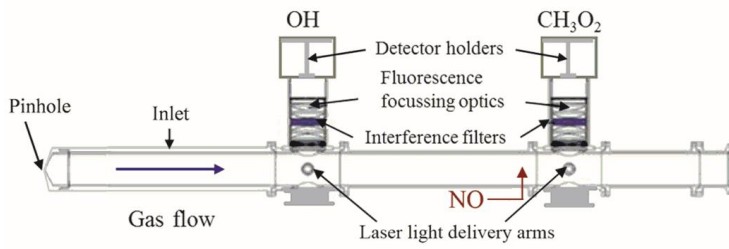

**Figure 1**. Vertical cross-section of the FAGE fluorescence cells. The first (left) fluorescence cell was used to detect OH fluorescence through a $(308.8 \pm 5.0)$ nm bandpass filter (transmission > 50 %) and the second cell to detect $CH_3O_2$ after titration with added NO to form $CH_3O$ using a bandpass filter between 320–430 nm with an average transmission > 80%.

Probe laser light was generated by a Nd:YAG (JDSU Q201-HD) pumped dye laser (SIRAH Credo-Dye-N) using a DCM

dye (Sirah) in ethanol and operating at 5 kHz pulse repetition frequency, with a pulse width at half maximum of 25 ns, typical pulse energy of 120 μJ pulse$^{-1}$ and a linewidth of 0.08 cm$^{-1}$ at 595 nm. The frequency doubled light at either ~308 nm (OH detection) or ~298 nm ($CH_3O$ detection), was focused into fibre optic cables to be delivered to the two detection cells. OH and $CH_3O$ radicals were separately detected by LIF spectroscopy by exciting at 307.99 nm using the $Q_1(2)$ rotational line of the $A^2\Sigma^+$ ($v' = 0$) ← $X^2\Pi_i$ ($v'' = 0$) OH transition in the first detection axis to monitor on-resonant fluorescence ($308.8 \pm 5.0$ nm)

and excitation at 297.79 nm in the $A^2A_1$ ($v'_3 = 3$) ← $X^2E$ ($v''_3 = 0$) $CH_3O$ transition in the second detection axis to monitor red-shifted off-resonant LIF (320-430 nm). Here $v_3$ refers to the C–O stretching vibrational mode of $CH_3O$ which demonstrates a progression in the LIF spectrum (Inoue et al., 1980;Kappert and Temps, 1989;Powers et al., 1997;Nagesh et al., 2014). The fluorescence in the two cells was collected orthogonal to the gas flow by two microchannel plate photomultiplier tubes (MCP-PMT) (Photek PMT325/Q/BI/G) equipped with a 50 ns gate unit (Photek GM10-50) for gated photon-counting, and the signal





was amplified using a pre-amplifier (Photek PA200-10). Further details on the OH detection and calibration in HIRAC have been reported previously (Winiberg et al., 2015).

The laser and photon-counting timing for $CH_3O$ detection was controlled by a delay pulse generator (9520 Quantum Composers). In order to avoid the scattered light from the probe laser damaging the MCP-PMT, the gate unit was opened 100 ns after the laser pulse to detect fluorescence integrated over a gate-width of 2 μs. The optimum gate-width of 2 μs (values in the range 1-3 μs were compared) is consistent with the $CH_3O$ fluorescence lifetimes, calculated to be in the range of $0.9 - 1.5$ μs, using the reported radiative lifetimes for $CH_3O$ of 1.5 μs (Inoue et al., 1979), 2.2 μs (Ebata et al., 1982) and $(4 \pm 2)$ μs (Wendt and Hunziker, 1979) and using the fluorescence quenching rate coefficients of $N_2$ and $O_2$ (Wantuck et al., 1987) to calculate the rate of quenching at the pressure in the FAGE detection cell $((2.65 \pm 0.05)$ Torr).

All LIF signals reported here were normalized to the probe laser power as measured with a laser power meter (Maestro, Gentec-EO) before the start of each LIF measurement. Fluctuations in the relative laser power were monitored via a photodiode (UDT-555UV, Laser Components) during the measurements and were accounted for in the signal normalization. The LIF spectrum was corrected for the laser-scattered background by subtracting the normalized offline signal recorded over 60 s at the end of each LIF measurement using an offline wavelength $\lambda$(offline = 300.29 nm) = $\lambda$(online = 297.79 nm) + 2.5 nm, well away from any $CH_3O$ absorption. Figure 2 shows the laser excitation spectrum centred at ~298 nm in the $\nu_3$ vibronic band recorded using an increment of $\Delta\lambda = 10^{-3}$ nm. The spectrum agrees well with previous work (Inoue et al., 1980;Kappert and Temps, 1989;Shannon et al., 2013). Methoxy radicals were generated via two methods, both using the 184.9 nm light generated by a Hg Pen-Ray lamp, namely the photolysis of methanol in nitrogen and the photolysis of water vapour in synthetic air (to generate OH) in the presence of methane to form $CH_3O_2$, followed by conversion to $CH_3O$ by added NO. Section 2.3 presents details on these two methods. Figure 3 shows typical laser excitation scans performed over a narrower range of wavelengths in order to locate $\lambda$(online) using the two methods for $CH_3O$ generation, and which show good agreement. There were no unexpected features in the LIF spectrum, consistent with no interference being anticipated in the FAGE measurements of $CH_3O$ as there were no other species in HIRAC absorbing at 298 nm and fluorescing at the wavelengths transmitted by the bandpass filter (average transmission > 80 % over $320 - 430$ nm).





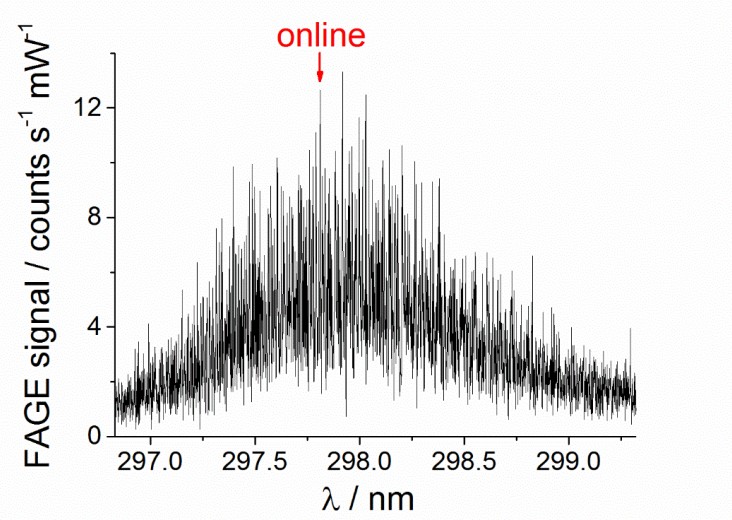

**Figure 2**. Laser excitation spectrum of the $A^2A_1$ ($v'_3 = 3$) ← $X^2E$ ($v''_3 = 0$) transition of the methoxy radical. $CH_3O$ radicals were obtained by photolysis of methanol in $N_2$ at 184.9 nm. Fluorescence cell pressure = (2.65 ± 0.05) Torr; wavelength increment $\Delta\lambda = 10^{-3}$ nm, with each point corresponding to 5000 laser shots. The red arrow indicates the wavelength $\lambda$(online) ~ 297.79 nm used for the time–resolved kinetic studies of $CH_3O$.

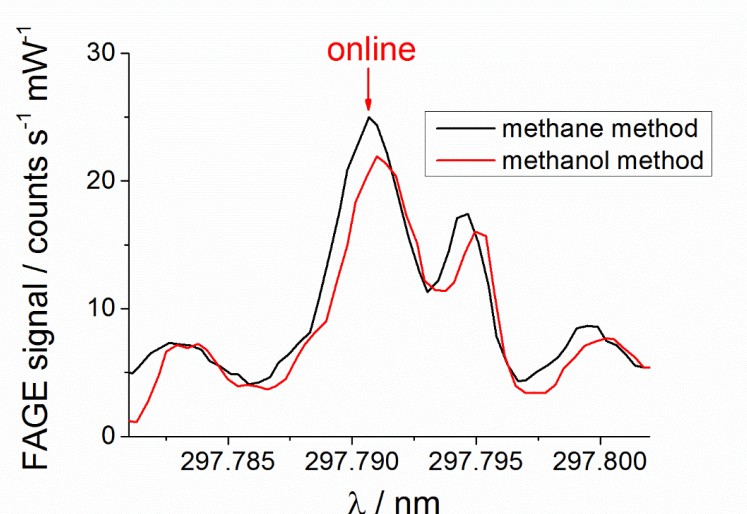

**Figure 3.** Typical laser excitation scans of $CH_3O$ performed over a much smaller range of wavelengths. Methoxy radicals were generated using $OH/CH_4$ (black line) to produce $5.5 \times 10^{10}$ molecule $cm^{-3}$ $CH_3O_2$, subsequently titrated to $CH_3O$ by adding NO, and the photolysis of methanol (red line) to generate $4.9 \times 10^{10}$ molecule $cm^{-3}$ $CH_3O$ directly. See main text for the description of the methods and calibration. The signal was normalised for the laser power ((10.3 ± 0.3) mW in methane method and (8.7 ± 0.2) mW in methanol method). Fluorescence cell pressure = (2.65 ± 0.05) Torr; wavelength increment $\Delta\lambda = 10^{-3}$ nm, with each point corresponding to 5000 laser shots. The red arrow indicates the wavelength $\lambda$(online) ~ 297.79 nm used for the time–resolved kinetic studies of $CH_3O$.




## 2.2 Optimisation of the NO concentration for methyl peroxy radical detection

As NO was added ~ 25 mm prior the methoxy detection axis (Fig. 1), some of the methoxy radicals formed by Reaction (R3) reacted further with NO before the fluorescence detection:

$$CH_3O + NO \rightarrow CH_2O + HNO \hspace{5cm} (R10)$$

$$CH_3O + NO + M \rightarrow CH_3ONO + M \, , \hspace{4cm} (R11)$$

where $M = N_2, O_2$. In addition to the above reactions, $CH_3O$ reacts with $O_2$ by Reaction (R4). Figure 4 shows the dependence
of the LIF signal on the concentration of NO obtained experimentally and by numerical simulations using Reactions (R3)–(R4) and (R10)–(R11) and outlined in the Supplementary Information. A maximum signal was obtained with added $[NO] = 6.7 \times 10^{13}$ molecule $cm^{-3}$ for a reaction time of 3 ms, estimated from the linear flow velocity within the FAGE reactor. Figure 4 shows that the functional dependence with added [NO] of the experimental $CH_3O$ signal and the simulated $[CH_3O]/[CH_3O_2]_0$ ratio display the same shape (within overlapping error limits) with the numerical simulations showing that $[CH_3O]/[CH_3O_2]_0$
at the detection axis was ~ 0.4 (i.e. 40 % conversion to $CH_3O$).

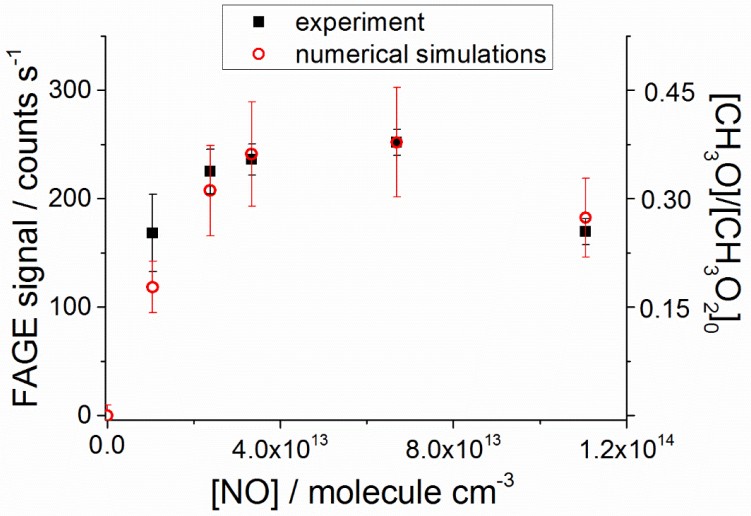

**Figure 4.** FAGE signal (left axis) and the ratio $[CH_3O]/[CH_3O_2]_0$ (right axis) as a function of the concentration of NO for a
reaction time of 3 ms. Black squares are experimental $CH_3O$ signals (errors are $1\sigma$) and red circles are the ratio $[CH_3O]/[CH_3O_2]_0$ generated by numerical simulations (percentage uncertainties are 20 %) using the chemistry system outlined in the main text and described in further detail in the Supplementary Information.

## 2.3 FAGE calibrations

$CH_3O$ and $CH_3O_2$ calibrations were carried out using the conventional radical source employed in fieldwork OH and $HO_2$ calibrations (Heard and Pilling, 2003) that produces radicals in a flow tube impinging just outside the FAGE inlet pinhole



(Winiberg et al., 2015) and is described in Sect. 2.3.1. Two methods of calibration have been used for $CH_3O_2$: the flow tube method and the kinetics of the self-reaction of $CH_3O_2$ carried out in HIRAC.

### 2.3.1 Calibration for methoxy radicals

In the $CH_3O$ calibration experiments nitrogen (BOC, > 99.998 %) was used as carrier gas. Part of the $N_2$ flow was passed through a methanol (Sigma Aldrich, ≥ 99.9 %) bubbler while the other portion bypassed the bubbler. The gas containing methanol vapour was then passed through a square cross-section flow tube of dimensions $13 \times 13$ (internal) $\times 300$ mm length with a flow rate of 40 slm (ensuring turbulent flow conditions), controlled by an electronic flow controller (Brooks, 0–100 slm air). The collimated light of a Hg Pen–Ray lamp (LOT–Oriel Hg–Ar) was directed across the flow tube (close to the downstream end) to photolyse methanol vapour. The flow tube output was impinged close to the FAGE inlet to sample $CH_3O$ radicals at atmospheric pressure through a 1 mm diameter pinhole (Fig. 1).

The concentration of $CH_3O$ radicals was calculated using Eq. (1):

$$[CH_3O] = [CH_3OH]\ \sigma_{CH3OH,\ 184.9\ nm}\ \Phi_{CH3O,\ 184.9\ nm}\ F_{184.9\ nm}\ \Delta t, \qquad (1)$$

where $\sigma_{CH3OH,\ 184.9\ nm}$ is the absorption cross section of methanol at 184.9 nm, $(6.35 \pm 0.28) \times 10^{-19}$ cm$^2$ molecule$^{-1}$, obtained by averaging reported values (Dillon et al., 2005; Jimenez et al., 2003; Nee et al., 1985), $F_{184.9\ nm}$ is the photon flux of 184.9 nm light and $\Delta t$ is the irradiation time of the gas. Although it is known, based on end-product analysis, that the scission of O–H bond is a major photolysis channel of methanol at 184.9 nm (Buenker et al., 1984; Porter and Noyes, 1959), the photodissociation quantum yield of $CH_3O$ at 184.9 nm, $\Phi_{CH3O,\ 184.9\ nm}$, has not been yet reported. Here it is assumed that $\Phi_{CH3O, 184.9\ nm}$ is equal to the photodissociation quantum yield at 193.3 nm, $\Phi_{CH3O,\ 193.3\ nm} = 0.86 \pm 0.10$, which has been reported (Satyapal et al., 1989). In order to determine the methanol vapour concentration in the flow tube, $[CH_3OH]$, separate experiments were carried out with the same calibration system to bubble deionised water instead of methanol with the same flow rate. The water vapour concentration, $[H_2O]$, was measured using a dew-point hygrometer (CR4, Buck Research Instrument) prior to the flow tube. Then $[CH_3OH]$ was calculated using the averaged $[H_2O]$ and the vapour pressures $p_{CH3OH}$ and $p_{H_2O}$ at the temperatures measured for $CH_3OH$ (13 °C) and $H_2O$ (15 °C) in the bubbler:

$$[CH_3OH]=[H_2O]\frac{p_{CH_3OH}}{p_{H_2O}} \qquad (2)$$

$N_2O$ photolysis at 184.9 nm to generate NO (via reaction of the photoproduct ($O^1D$) with $N_2O$ giving a known yield of NO), which was subsequently measured using a commercial analyser, was used as a chemical actinometer to obtain the product $F_{184.9\ nm} \times \Delta t$ (Winiberg et al., 2015) and hence calculate $[CH_3O]$ via Eq. (1). The photolysis time, $\Delta t$, was estimated to be 8.3 ms, using the volumetric flow rate and the geometric parameters of the flow tube (assuming plug flow) and was in turn used to determine $F_{184.9\ nm}$. Although it is the product $F_{184.9\ nm} \times \Delta t$ which is used to calculate $[CH_3O]$, any change in the volumetric flow rate between the calibration and actinometry experiments will change $\Delta t$, and hence the product was corrected for any changes in volumetric flow rate. A range of $[CH_3O]$ at constant $[CH_3OH]$ was produced by changing the electrical current through the Hg lamp between 0 and 20 mA, and hence $F_{184.9\ nm}$, to generate the calibration plot presented in Fig. 5.





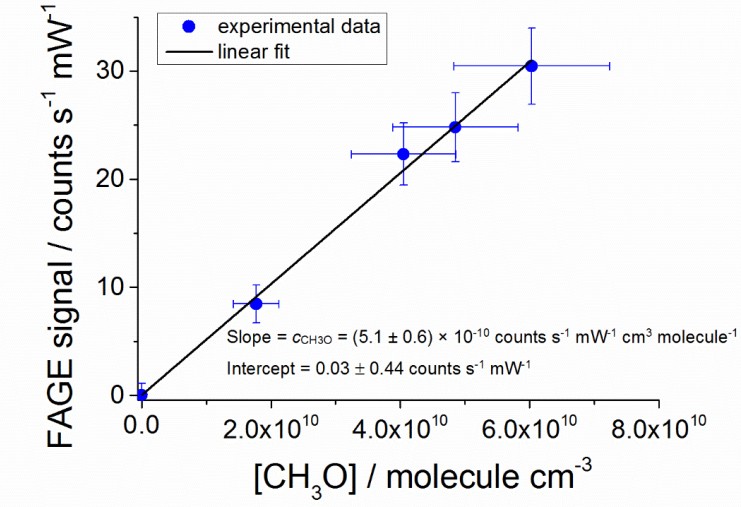

**Figure 5**. FAGE calibration for $CH_3O$ at atmospheric pressure and 293 K; laser power $P = (12.9 \pm 0.3)$ mW and pressure in the detection cell of $(2.65 \pm 0.05)$ Torr. The FAGE signal, including the measurement with the Hg lamp turned off ($[CH_3O] = 0$), was obtained after subtraction of the offline signal, $(12.3 \pm 0.9)$ counts $s^{-1}$ $mW^{-1}$. Averaging time per point

= 120 s. The error limits in $[CH_3O]$ and the FAGE signal for the $x$ and $y$ axes respectively are representative of the $1\sigma$ overall uncertainty, which contains the total systematic and statistical errors (see text for details of these). The error limits shown in the legend are the standard errors in the slope and intercept of the fit to the experimental data.

**2.3.2 Calibration for methyl peroxy radicals**

**2.3.2.1 Flow tube method**

Methyl peroxy radicals were generated by water photolysis at 184.9 nm (Reaction (R7)) to give OH followed by the reaction with excess methane in air (BOC, synthetic BTCA 178) – Reactions (R1)–(R2) to give $CH_3O_2$. The calibrations were performed using the set–up described above. Methane (BOC, CP grade, 99.5 %) was flowed at 82.5 sccm to convert OH into

$CH_3$, which subsequently reacted rapidly with $O_2$ to form $CH_3O_2$. Figure S1 (Supplementary Information) shows an example of the OH signal with and without $CH_4$. The signal in the presence of $CH_4$ was $(0.04 \pm 0.04)$ of the signal in the absence of $CH_4$ showing that $(0.96 \pm 0.04)$ of OH was converted into $CH_3O_2$. The result is in agreement with the estimation of the fraction of OH titrated to $CH_3O_2$, 0.97, using a rate coefficient of $6.4 \times 10^{-15}$ $cm^3$ $molecule^{-1}$ $s^{-1}$ for the OH + $CH_4$ reaction (Atkinson et al., 2006) and an average residence time of OH in the calibration flow tube of 11 ms determined using the volumetric flow

rate and the geometric parameters of the flow tube and position of the Hg pen lamp.

The concentration of $CH_3O_2$ was determined using Eq. (3):

$$[CH_3O_2] = 0.96\,[OH] = 0.96\,[H_2O]\ \sigma_{H2O,\,184.9\,nm}\ \Phi_{H2O,\,184.9\,nm}\ F_{184.9\,nm}\ \Delta t \tag{3}$$

where $\sigma_{H2O,\,184.9\,nm}$ is the absorption cross section of water vapour at 184.9 nm, $(7.22 \pm 0.22) \times 10^{-20}$ $cm^2$ $molecule^{-1}$ (Cantrell et al., 1997;Creasey et al., 2000) and $\Phi_{H2O,\,184.9\,nm}$ is the photodissociation quantum yield of OH, which is equal to unity. The values of $F_{184.9\,nm}$ and $\Delta t$ were determined as described in the Sect. 2.3.1. No loss of $CH_3O_2$ by reaction with the $HO_2$ radicals generated by the reaction of H atoms with $O_2$ (R8) was encountered over the residence time of the radicals in the calibration



flow tube (~11 ms) as $CH_3O_2$ reacts with $HO_2$ on a ten second timescale as determined using a reaction rate coefficient of 5.2 $\times$ $10^{-12}$ $cm^3$ $molecule^{-1}$ $s^{-1}$ (Atkinson et al., 2006) and the radical concentrations in the flow tube. The $CH_3O_2$ radicals sampled through the FAGE pinhole expansion to a pressure of 2.65 Torr reached the detection region in about 85 ms while the calculated $CH_3O_2 + HO_2$ reaction half-life at this reduced pressure in the FAGE inlet was thousands of seconds and any change in the $CH_3O_2$ concentration is expected to be negligible.

Figure 6 shows results obtained from three separate calibration experiments. In the first two experiments air was humidified by passing a fraction of the air flow (40 slm total flow rate) through a deionised water bubbler. The hygrometer measured $7.5 \times 10^{16}$ molecule $cm^{-3}$ of water vapour prior to the calibration flow tube and the concentration of methane in the flow tube was $5 \times 10^{16}$ molecule $cm^{-3}$. In the second experiment, a series of FAGE measurements were performed using a photon flux

of ~ $1.6 \times 10^{14}$ photon $cm^{-2}$ $s^{-1}$ to generate ~ $4.5 \times 10^9$ molecule $cm^{-3}$ $CH_3O_2$. In the third experiment $[CH_4] = 10^{17}$ molecule $cm^{-3}$ and all the air flow (now at 20 slm) was passed through the water bubbler to obtain $3 \times 10^{17}$ molecule $cm^{-3}$ $H_2O$ vapour. The concentration of $CH_3O_2$ was varied by changing the photon flux in the range of $0–1.5 \times 10^{14}$ photon $cm^{-2}$ $s^{-1}$ to generate $[CH_3O_2] = 1.5–4.5 \times 10^{10}$ molecule $cm^{-3}$.

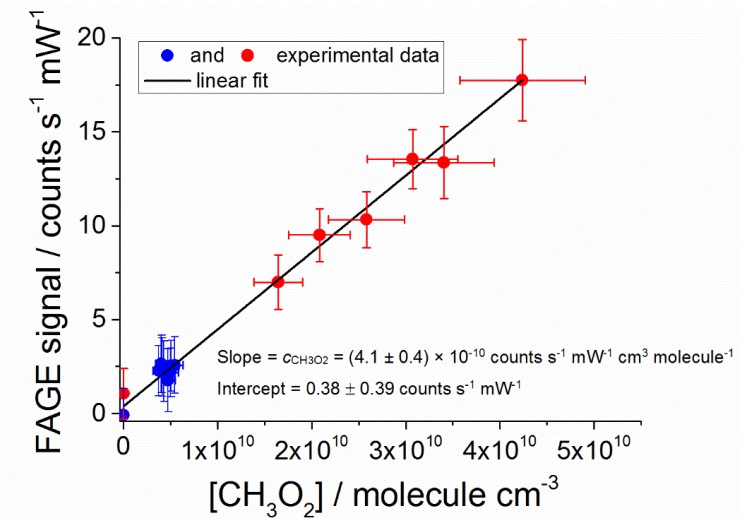

Figure 6. FAGE calibration for $CH_3O_2$ at atmospheric pressure and 293 K. The data were obtained by three separate experiments: two of them generating $[CH_3O_2] \cong 4.5 \times 10^9$ molecule $cm^{-3}$ in the calibration flow tube (blue circles); laser power

$P = (9.5 \pm 0.3)$ mW and $(11.6 \pm 0.4)$ mW, respectively and one experiment using $[CH_3O_2]$ in the range of $1.5–4.5 \times 10^{10}$ molecule $cm^{-3}$ (red circles); $P = (9.2 \pm 0.2)$ mW. The pressure in the FAGE detection cell was maintained $(2.65 \pm 0.05)$ Torr in all experiments. Averaging time per point = 120 s. The error limits in $[CH_3O_2]$ and the FAGE signal for the $x$ and $y$ axes respectively are representative for the $1\sigma$ overall uncertainty, which contains the total systematic and statistical errors. The error limits shown in the legend are the standard errors in the slope and intercept of the fit to the experimental data.


### 2.3.2.2 $CH_3O_2$ second–order decay method

The principle behind this calibration method is that the second–order decay of $CH_3O_2$ is dependent upon its initial concentration, and hence its quantification offers an alternative way to calibrate the signal. The experiments were performed





in the HIRAC chamber at 295 K and 1 bar of synthetic air obtained by mixing high purity oxygen (BOC, > 99.999 %)) and nitrogen (BOC, > 99.998 %) in the ratio of $O_2$:$N_2$ = 1:4. Methane (BOC, CP grade, $2-3 \times 10^{17}$ molecule $cm^{-3}$) and molecular chlorine (Sigma Aldrich, $\geq$ 99.5 %, $0.3-2.1 \times 10^{14}$ molecule $cm^{-3}$) were delivered to the chamber. Eight UV black lamps (Phillips, TL–D 36W/BLB, $\lambda$ = 350–400 nm) housed in quartz tubes mounted radially inside the reactive volume were used

to photolyse $Cl_2$ to generate Cl atoms and initiate the chemistry:

$$CH_4 + Cl \rightarrow CH_3 + HCl \tag{R12}$$
$$CH_3 + O_2 + M \rightarrow CH_3O_2 \ + M \tag{R2}$$

Numerical simulations using the chemical system described in Table S3 in the Supplementary Information showed that $[Cl]_0$ = $1-6 \times 10^6$ molecule $cm^{-3}$ (varied by changing the initial $[Cl_2]$). The high excess of methane ($2-3 \times 10^{17}$ molecule $cm^{-3}$) relative to $[Cl]_0$ ensured that the reactions of the Cl atoms with the self–reaction products formaldehyde and methanol were negligible. In each HIRAC experiment the lamps were alternatively turned on for 2–3 min and then off over 1–2 min to generate a series of typically 3–4 $CH_3O_2$ kinetic decays.

In order to detect $CH_3O_2$ the FAGE instrument was coupled to HIRAC through a custom-made ISO-K160 flange to sample the gas with a flow rate of ~ 3 slm. For most measurements, the 1 mm pinhole of the 280 mm long FAGE inlet was sampling ~230 mm from the chamber wall as in the OH measurements reported previously (Winiberg et al., 2015). Additional investigations into any $CH_3O_2$ gradient across the ~ 600 mm radius of HIRAC were conducted using measurements of $CH_3O_2$ formed by the $CH_4$ reaction with $O(^1D)$ generated by the photolysis of $O_3$ at 254 nm followed by the reaction of the produced

$CH_3$ radical with $O_2$ at 295 K and 1 bar of synthetic air. An extended FAGE inlet (length 520 mm) was used to sample along 500 mm across the chamber starting with the inlet pinhole flush at the wall. A constant concentration of $CH_3O_2$ was found (within the 10 % overall error of the measurement) for all the sampled distance 0 – 500 mm from the wall (note that 0 mm here refers to the FAGE inlet being at an equivalent position to the wall away from the mounting flange). The absence of a $CH_3O_2$ gradient across the chamber provides evidence of the efficacy of the mixing in HIRAC and shows that the wall–loss of

$CH_3O_2$ is negligible and hence that a shorter inlet, and hence distance from inlet to $CH_3O_2$ detection axis could be used in future $CH_3O_2$ FAGE measurements within HIRAC, improving further the sensitivity.

**2.4 Methoxy radical measurements within HIRAC**

The experiment was carried out in HIRAC at 295 K and 1 bar of $N_2$ (BOC, > 99.998 %), but without any NO added to the FAGE cell (the cell furthest from the pinhole as shown in Fig. 1) so that $[CH_3O]$ is measured directly. Initial concentrations in

HIRAC were: $[CH_4]_0$ = $4.50 \times 10^{17}$ molecule $cm^{-3}$ and $[Cl_2]_0$ = $5.57 \times 10^{15}$ molecule $cm^{-3}$. After adding the reagents into the chamber the lamps (*vide supra*) were turned on to generate $CH_3O$ by Reaction (R6.b).

**3 Results**

**3.1 Sensitivity and detection limits for $CH_3O_2$ and $CH_3O$ radicals obtained from calibrations**

**3.1.1 Flow tube method**

The FAGE sensitivity for $CH_3O_2$ ($C_{CH3O2}$) and $CH_3O$ ($C_{CH3O}$), is the slope of the linear regressions in Fig. 5 and Fig. 6, which were $C_{CH3O2}$ = $(4.1 \pm 1.4) \times 10^{-10}$ counts $cm^3$ molecule$^{-1}$ s$^{-1}$ mW$^{-1}$ and $C_{CH3O}$ = $(5.1 \pm 2.2) \times 10^{-10}$ counts $cm^3$ molecule$^{-1}$ s$^{-1}$ mW$^{-1}$. The error limits, 34 % for $C_{CH3O2}$ and 43 % for $C_{CH3O}$, are overall $2\sigma$ uncertainties calculated using the sum in quadrature of the systematic uncertainties, 33 % for $CH_3O_2$ and 42 % for $CH_3O$ (details in Section 3.2.1), and the statistical errors from the calibration plots, ~ 8 %. The higher errors in $C_{CH3O}$ compared to $C_{CH3O2}$ are due to the uncertainty





in the methanol concentration, which is not determined directly (*vide supra*), $1\sigma \cong 7\ \%$ and the error in the yield of $CH_3O$. The value of the $CH_3O$ photolysis yield from $CH_3OH$ reported at 193 nm was used ($0.86 \pm 0.10$), which has an uncertainty of 11.63 % at the $1\sigma$ level (Satyapal et al., 1989).

From the sensitivity factor, $C$, the limit of detection (*LOD*) was calculated using Eq. (4) and assuming Poisson statistics
appropriate for single photon counting:

$$LOD(CH_3O_2)= \frac{S/N}{C_{CH_3O_2}P}\sqrt{\frac{BKG}{t}\left(\frac{1}{m}+\frac{1}{n}\right)}, \qquad\qquad LOD(CH_3O)= \frac{S/N}{C_{CH_3O}P}\sqrt{\frac{BKG}{t}\left(\frac{1}{m}+\frac{1}{n}\right)}, \qquad (4)$$

where $S/N$ is the signal-to-noise ratio, $P$ is the laser power, $BKG$ is the background signal due to laser scatter, scattered visible
light and the detector dark counts, and had a typical value of $\sim$100 counts s$^{-1}$, $t$ is the time per data point, $m$ represents the number of online data points and $n$ is the number of offline data points. For a typical 5 min averaged signal, i.e. $m = n = 150$, $S/N = 2$, $P = 15$ mW and $t = 1$ s, and using the values of $C$ from the calibration, $LOD(CH_3O_2) = 3.8 \times 10^8$ molecule cm$^{-3}$ and $LOD(CH_3O) = 3.0 \times 10^8$ molecule cm$^{-3}$. An increase of the averaging time to 1 hour, i.e. $m = n = 1800$ data points, results in a decrease of the detection limits to $LOD(CH_3O_2) = 1.1 \times 10^8$ molecule cm$^{-3}$ and $LOD(CH_3O) = 8.7 \times 10^7$ molecule cm$^{-3}$.

Although $CH_3O_2$ has not been measured specifically in the atmosphere, there have been several calculations of its concentration using numerical models. In general, the concentration of $CH_3O_2$ is a function both of the loadings of volatile organic compounds (VOCs) and the levels of $NO_x$. For the clean, remote environments at Cape Verde in the tropical Atlantic ocean and in the Borneo rainforest $[CH_3O_2]$ is calculated to peak around $6 \times 10^8$ molecule cm$^{-3}$ and about $2 \times 10^8$ molecule cm$^{-3}$, respectively at noon using the modeling studies reported by Whalley et al. (Whalley et al., 2010;Whalley et al., 2011).

Therefore, it should be possible using the FAGE conversion method to $CH_3O$ and for an averaging time of 1 hour (*vide supra*) to achieve a measurement of atmospheric levels of $CH_3O_2$ in such clean environments, and shorter averaging times in some cases. Further optimizations of FAGE sensitivity can be achieved by the removal of the fibre optic cables to deliver the probe laser beam directly to the $CH_3O$ detection cell to increase the laser power and, by increasing the pulse repetition frequency above the current value of 5 kHz (but without significant reduction in the pulse energy). The present investigations into the
change of sensitivity with pressure in the range from 2.65–10.00 Torr found that 2.65 Torr is the optimum value in this pressure interval. Hence an additional improvement in the sensitivity might be obtained by using a lower detection cell pressure than the current value of 2.65 Torr using a more powerful pump. It should be also noted that the distance from the inlet pinhole to the laser–axis in the $CH_3O$ and $CH_3O_2$ fluorescence cell (Figure 1, $\sim$ 580 mm) is considerably longer than the corresponding distance in the ground–based field fluorescence cell for OH and $HO_2$ detection (88 mm), and improvements in sensitivity
would be expected for a shorter pinhole–to–laser excitation distance for $CH_3O_2$. The further optimizations of sensitivity could potentially enable $CH_3O_2$ measurements to be made in urban environments where $CH_3O_2$ concentrations are estimated to be considerably lower, for example a few $10^7$ molecule cm$^{-3}$ based on modeling results (Whalley et al., to be submitted).

### 3.1.2 Methyl peroxy calibration using kinetics of the $CH_3O_2$ second–order decay

An alternative method of calibration for $CH_3O_2$ was to generate $CH_3O_2$ radicals in HIRAC to monitor the temporal decay of
the $CH_3O_2$ FAGE signal once the photolysis lamps were turned off. Figure 7 shows an example of a decay in the $CH_3O_2$ signal generated by extinguishing the HIRAC lamps following the production of $CH_3O_2$ by the Cl atom initiated oxidation of $CH_4$ in the presence of $O_2$ (Reactions (R12) and (R2)). In the absence of other processes, the loss of $CH_3O_2$ is described by the integrated second–order rate law equation describing the $CH_3O_2$ self–reaction (Reaction (R6)):

$$\frac{1}{[CH_3O_2]_t}=\frac{1}{[CH_3O_2]_0}+2 \cdot k_{obs}t, \qquad (5)$$



where $[CH_3O_2]_t$ is the methyl peroxy concentration at reaction time $t$, $[CH_3O_2]_0$ is the initial concentration when the lights are switched off and $k_{obs}$ is the observed rate coefficient (which is not equal to $k_6$, see below). Using $[CH_3O_2] = \frac{S_{CH_3O_2}}{C_{CH_3O_2}}$, where $S_{CH3O2}$ is the signal measured by FAGE and $C_{CH3O2}$ is the instrument sensitivity, Eq. (6) is obtained for the temporal profile of the methyl peroxy signal:

$$\frac{1}{(S_{CH_3O_2})_t} = \frac{1}{(S_{CH_3O_2})_0} + \frac{2 \cdot k_{obs}t}{C_{CH_3O_2}} \qquad \text{or} \qquad (S_{CH_3O_2})_t = 1 / \left( \frac{1}{(S_{CH_3O_2})_0} + \frac{2 \cdot k_{obs}t}{C_{CH_3O_2}} \right), \qquad (6)$$

In Eq. (6) $(S_{CH3O2})_t$ and $(S_{CH3O2})_0$ are the signal at time $t$ and $t = 0$ respectively.

10    Eq. (6) was fitted to the experimental decays of $S_{CH3O2}$ (see Fig. 7 as an example) fixing $k_{obs}$ to the IUPAC recommendation, $k_{obs} = (4.8 \pm 1.1) \times 10^{-13}$ cm$^3$ molecule$^{-1}$ s$^{-1}$, in order to obtain $C_{CH3O2}$. Eighteen $CH_3O_2$ decays were analysed, which yielded an average value of $C_{CH3O2} = (5.6 \pm 1.7) \times 10^{-10}$ counts cm$^3$ molecule$^{-1}$ s$^{-1}$ mW$^{-1}$. The error limit, 30 %, is the $2\sigma$ composite error calculated as the sum in quadrature of the total systematic uncertainty, 29 % (see Section 3.2.2), and the average random error of all determinations, with 8 %, taken as two standard errors in the fit of Eq. (6) to the $CH_3O_2$ temporal decays. This value agrees well with $C_{CH3O2} = (4.1 \pm 1.4) \times 10^{-10}$ counts cm$^3$ molecule$^{-1}$ s$^{-1}$ mW$^{-1}$ obtained from the flow-tube calibration method

15    (section 3.1.1).

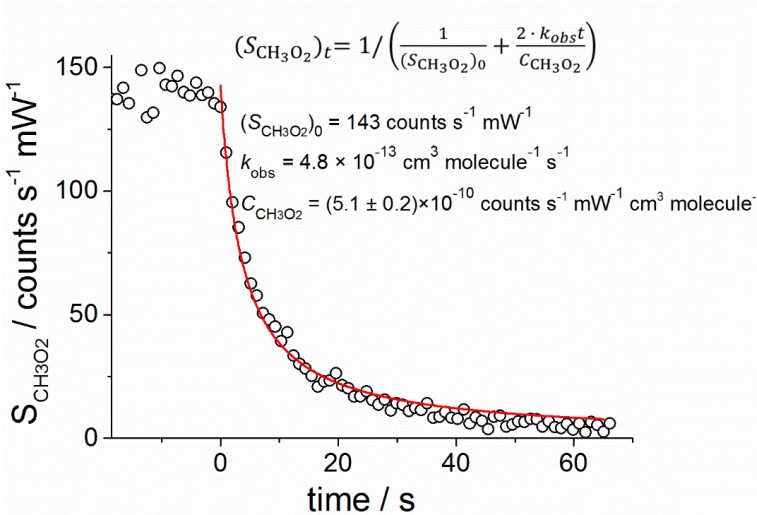

**Figure 7**. Second–order decay of the normalized $CH_3O_2$ signal with 1 second time resolution generated in HIRAC using
Cl/CH$_4$/O$_2$ and black lamps (see main text for details); $[CH_4]_0 = 2.3 \times 10^{17}$ molecule cm$^{-3}$ and $[Cl_2]_0 = 5.8 \times 10^{13}$ molecule cm$^{-1}$ at 295 K and 1 bar mixture of N$_2$:O$_2$ = 4:1. At time zero the lamps were turned off. Fitting Eq. (6) to the data yielded $C_{CH3O2} = (5.1 \pm 0.2) \times 10^{-10}$ counts cm$^3$ molecule$^{-1}$ s$^{-1}$ mW$^{-1}$ (statistical error at $1\sigma$ level).

25    Based on the lack of a measurable $CH_3O_2$ radical gradient across HIRAC (Section 2.3.2.2, *vide supra*) it is assumed that the loss of $CH_3O_2$ to the walls of HIRAC in these experiments was negligible over the timescale of 1–2 min of the temporal decay measurements. Our finding is consistent with previous results showing that the heterogeneous wall–loss rates for $CH_3O_2$ were significantly lower than the corresponding removal rates of HO$_2$ (Miyazaki et al., 2010;Mihele et al., 1999;Fuchs et al.,




2008). Using a 30 cm long glass tube of 2 cm diameter, Miyazaki et al. measured that the heterogeneous removal efficiency for $CH_3O_2$ was six times lower than for $HO_2$. The $HO_2$ wall–loss rate coefficient at room temperature and 1000 mbar in HIRAC was found to be of ~ $10^{-2}$ s$^{-1}$ (Winiberg et al., 2015). Therefore, it can be expected that the wall–loss rate coefficient of $CH_3O_2$ in HIRAC was $k_{loss} \cong 10^{-3}$ s$^{-1}$ and so is not considered in the analysis here for $CH_3O_2$ decays which typically last for ~ 100 s.

In order to investigate the sensitivity of $C_{CH3O2}$ obtained by the kinetic analysis of the $CH_3O_2$ decay to $k_{loss}$ higher than $10^{-3}$ s$^{-1}$, a wall–loss rate coefficient of $10^{-2}$ s$^{-1}$ was included in the analysis of the experimental decays of $CH_3O_2$ to obtain $C_{CH3O2}$, but only an increase in $C_{CH3O2}$ of about 6 % on average was seen. A small deviation of the experimental data from the fit was obtained at the end of the measurements whether or not $k_{loss}$ was included in the analysis (Fig. 7). The role of potential secondary chemistry at later times of the reaction will be investigated in future kinetic studies of the $CH_3O_2$ self–reaction.

Using the average sensitivity factor $C_{CH3O2} = (5.6 \pm 1.7) \times 10^{-10}$ counts cm$^3$ molecule$^{-1}$ s$^{-1}$ mW$^{-1}$ determined by the $CH_3O_2$ decay method in HIRAC, and for a signal to noise ratio $S/N = 2$, a laser power $P = 15$ mW and a time per data point $t = 1$ s in Eq. (4) results in an improved (compared with the flow tube calibration) $LOD(CH_3O_2) = 2.8 \times 10^8$ molecule cm$^{-3}$ for 5 min averaging time, i.e. 150 online data points ($m$) and 150 offline points ($n$). The corresponding $LOD$ for an averaging time of 1 hour, i.e. $m = n = 1800$ is $LOD(CH_3O_2) = 7.9 \times 10^7$ molecule cm$^{-3}$.

It should be noted that the observed rate coefficient, $k_{obs}$, is larger than the second–order rate coefficient of just the $CH_3O_2$ recombination reaction (R6), $k_6$, as the methoxy radicals generated by channel R6.b react rapidly with molecular oxygen present in large excess, $5 \times 10^{18}$ molecule cm$^{-3}$, to produce $HO_2$ (R4) which in turn reacts with $CH_3O_2$ (R13):

$$CH_3O_2 + CH_3O_2 \rightarrow CH_3OH + CH_2O + O_2 \tag{R6.a}$$
$$CH_3O_2 + CH_3O_2 \rightarrow CH_3O + CH_3O + O_2 \tag{R6.b}$$
$$CH_3O + O_2 \rightarrow CH_2O + HO_2 \tag{R4}$$
$$CH_3O_2 + HO_2 \rightarrow CH_3OOH + O_2 \tag{R13.a}$$
$$CH_3O_2 + HO_2 \rightarrow CH_2O + O_2 + H_2O \tag{R13.b}$$

As each $HO_2$ radical consumes one $CH_3O_2$ species (R13) on the time scale of the reaction (R6), $k_{obs}$ is given by (Sander and Watson, 1981;Lightfoot et al., 1990):

$$k_{obs} = k_6 \cdot (1 + r_{6.b}) \tag{7}$$

where $r_{6.b}$ is the branching ratio for the reaction channel R6.b. According to IUPAC (Atkinson et al., 2006), there is a 23 % uncertainty in $k_{obs}$ of the $CH_3O_2$ recombination at 298 K with a recommended value $k_{obs} = 4.8 \times 10^{-13}$ cm$^3$ molecule$^{-1}$ s$^{-1}$. This value corresponds to $k_6 = (3.5 \pm 1.0) \times 10^{-13}$ cm$^3$ molecule$^{-1}$ s$^{-1}$ and $r_{6.b} = 0.37 \pm 0.06$ (Atkinson et al., 2006).

       In order to check the validity of Eq. (7) in the presence of $HO_2$ removal by self–reaction and wall–loss, numerical simulations were performed to generate $CH_3O_2$ decays using a system incorporating the chemistry described by Reactions
(R4), (R6), R(13) and R(14) (*vide infra*) and a heterogeneous loss of $HO_2$, $k_{loss(HO2)}$ (Supplementary Information). The rate coefficients were sourced from the IUPAC preferred values at 298 K (Table S3 in Supplementary Information) and $k_{loss(HO2)}$ was varied. The simulated decays of $[CH_3O_2]$ vs. time were analysed using Eq. (5) (see Fig. S3 as an example) and gave an average observed rate coefficient of $k_{obs} = 4.7 \times 10^{-13}$ cm$^3$ molecule$^{-1}$ s$^{-1}$, which is only 2 % lower than the IUPAC recommendation, for $k_{loss(HO2)}$ varied between 0.01–0.10 s$^{-1}$ and, hence confirm the applicability of Eq. (7).

$$HO_2 + HO_2 \rightarrow H_2O_2 + O_2 \tag{R14.a}$$
$$HO_2 + HO_2 + M \rightarrow H_2O_2 + O_2 + M \tag{R14.b}$$



### 3.2 Calibration uncertainties

#### 3.2.1 Flow tube method

The $2\sigma$ error associated with $C_{CH3O2}$ of 34 % obtained by the flow tube method ($C_{CH3O2}$ = (4.1 ± 1.4) × $10^{-10}$ counts $cm^3$

molecule$^{-1}$ s$^{-1}$ mW$^{-1}$), represents the overall uncertainty calculated using the sum in quadrature of the systematic uncertainty, 33 %, and the statistical error from the calibration plots, ~ 8 %. The overall 34 % uncertainty is similar to the previous estimated total uncertainty, 36 %, in the use of the same method for calibration of OH and $HO_2$ measurements in HIRAC, where no $CH_4$ is added (Winiberg et al., 2015). The flow tube method is a proven method to generate known amounts of OH and $HO_2$ by the photolysis of $H_2O$ at 184.9 nm in order to calibrate field instruments (Heard and Pilling, 2003).

The largest contribution to the total error of the method came from the 28 % total uncertainty in the photon flux of the calibration source, $F_{184.9nm}$. The product $F_{184.9\ nm} \times \Delta t$ is determined using $N_2O$ actinometry relying on the measurement of [NO] in trace amounts (0.2–1.5 ppbv) using a commercial NO analyser (Thermo Electron Corporation 42C) followed by the data analysis using four rate coefficients each with ~ 10 % uncertainty (Burkholder et al., 2015). Although it is the product $F_{184.9\ nm} \times \Delta t$ which is directly determined by the actinometric method and used to calculate the concentration of radicals to

calibrate FAGE (Eq. (3)), any difference in the volumetric flow rate between the calibration and actinometry experiments will change $\Delta t$. Therefore, the uncertainty in $\Delta t$, 2 %, needs to be accounted for. The contributions from the rest of the terms in Eq. (3) to the systematic uncertainty in the determination of [$CH_3O_2$] by this method were as follows: 6 % total error in $\sigma_{H2O,\ 184.9nm}$ (Cantrell et al., 1997), 10 % uncertainty in [$H_2O$], taken from the instrumental uncertainty of the hygrometer and 4 % error in the yield of $CH_3O_2$ produced by the OH conversion into $CH_3$ followed by the $CH_3 + O_2$ reaction. The contribution of the

uncertainties in the FAGE measurements to the 33 % overall systematic uncertainty in the calibration were estimated to consist of 12 % in the online FAGE signal and 6 % uncertainty in the laser power measured by the laser power meter and used to normalize the data. The uncertainty associated with the online signal, 6 % at $1\sigma$ level, was calculated as the average deviation of the signal value due to the error limits of ± 5 × $10^{-4}$ nm in the online wavelength position (see the typical laser excitation scans shown in Fig. 3).

#### 3.2.2 $CH_3O_2$ second–order decay calibration

The largest contribution to the calculated overall $2\sigma$ uncertainty of 30 % in $C_{CH3O2}$ obtained by the $CH_3O_2$ second–order decay method ($C_{CH3O2}$ = (5.6 ± 1.7) × $10^{-10}$ counts $cm^3$ molecule$^{-1}$ s$^{-1}$ mW$^{-1}$), derives from the 23 % error in the IUPAC preferred value of the observed rate coefficient for the effective $CH_3O_2$ self–reaction, $k_{obs}$ = (4.8 ± 1.1) × $10^{-13}$ $cm^3$ molecule$^{-1}$ s$^{-1}$ (Atkinson et al., 2006). It is instructive to examine the origin of the 23 % error. The studies which led to the IUPAC

recommendation utilized the UV–absorption of $CH_3O_2$, typically at 250 nm, and the determined quantity was the ratio between the observed rate coefficient and the absorption cross section of $CH_3O_2$, $k_{obs}/\sigma_{250nm}$. IUPAC and the Jet Propulsion Laboratory (JPL) recommend 3.9 × $10^{-18}$ and 3.8 × $10^{-18}$ $cm^2$ molecule$^{-1}$, respectively for $\sigma_{250nm}$ (Atkinson et al., 2006;Burkholder et al., 2015). The JPL recommendation (Burkholder et al., 2015) is the cross section obtained by the re–evaluation of the previous reported UV–absorption spectra by Tyndall et al. in 2001 (Tyndall et al., 2001), yielding $\sigma_{250nm}$ = 3.78 × $10^{-18}$ $cm^2$ molecule$^{-1}$.

Tyndall et al. used a cross section of (4.26 ± 0.52) × $10^{-18}$ $cm^2$ molecule$^{-1}$ (error at $2\sigma$) for the maximum at 237.3 nm, obtained by analysing the shape of the absorption spectra between 200–300 nm reported since 1990. The studies before 1990 were not included due to errors in the calibration of the $CH_3O_2$ cross section leading to large discrepancies in the reported values. The 2001 evaluation of Tyndall et al. calculated $k_{obs}$ = (4.7 ± 0.8) × $10^{-13}$ $cm^3$ molecule$^{-1}$ s$^{-1}$ where the error limits are two standard deviations of the mean. Including an error of 10 % in the cross section of $CH_3O_2$, as suggested by the authors (Tyndall et al.,



2001), a 19 % composite uncertainty in $k_{obs}$ is obtained. The result is in good agreement with the 23 % uncertainty in the IUPAC recommendation.

The remaining contributions to the uncertainty in the calibration using the $CH_3O_2$ second–order decay method are: 6 % error in the laser power, 12 % uncertainty in the online signal determined by how well the laser is able to find the online
wavelength position (*vide supra*) and 10 % error in $(S_{CH3O2})_0$ in Eq. (6), the value of the $CH_3O_2$ signal at the moment when the HIRAC lamps were turned off to generate a second–order decay.

### 3.2.3 Comparison between the FAGE sensitivities for CH₃O₂ obtained by the two calibration methods

The FAGE sensitivity factor obtained using the flow tube method, $C_{CH3O2} = (4.1 \pm 1.4) \times 10^{-10}$ counts $cm^3$ molecule$^{-1}$ s$^{-1}$ mW$^{-1}$, is 27 % lower but has overlapping error limits with the result found using the $CH_3O_2$ second–order decay method, $C_{CH3O2} =$
$(5.6 \pm 1.7) \times 10^{-10}$ counts $cm^3$ molecule$^{-1}$ s$^{-1}$ mW$^{-1}$ (uncertainties quoted to $2\sigma$). The calculated overall error in the $CH_3O_2$ second–order decay method, 30 %, is similar to the total uncertainty in the flow tube method, 34 %. The flow tube method is known to reliably generate accurate concentrations of radicals and has been used for many years in the calibration of FAGE instruments employed in field measurements of OH and $HO_2$ (Heard and Pilling, 2003). The flow tube method has also been validated by using alternate methods of calibration, for example using the decay of a hydrocarbon in the HIRAC chamber to
obtain [OH] (Winiberg et al., 2015). The method of using a time–resolved kinetic quantity to derive a calibration factor was validated for $HO_2$ in HIRAC, where $C_{HO2}$ obtained from analysis of the temporal decay of $HO_2$ agreed with $C_{HO2}$ from the flow tube method (Winiberg et al., 2015). These results suggests that the sensitivity of the FAGE system, represented by the value of $C$, is not changed between sampling from the calibration flow tube and sampling from within HIRAC itself.

The accuracy of the $CH_3O_2$ temporal decay method is largely determined by the accuracy of $k_{obs}$ (see section 3.2.2. above).
The quantity measured in the previous kinetic studies of $CH_3O_2 + CH_3O_2$ is $k_{obs}/\sigma_{250nm}$ and hence the accuracy of $k_{obs}$ is directly affected by any systematic errors in the determination of $\sigma_{250nm}$. In order to make $C_{CH3O2}$ derived from the temporal decay and flow tube methods of the same, the value of $k_{obs}$ would need to be reduced by ~ 25 %, which in turn requires a ~ 25 % reduction in $\sigma_{250nm}$. It is noted that the UV–absorption spectrum of $CH_3O_2$ is relatively broad and hence may prevent a selective detection due to the difficulty to discriminate from the potential presence of other species also absorbing around 250 nm, such as $Cl_2$
and $CH_3CHO$ used in concentrations as high as $10^{16}$ molecule cm$^{-3}$, while [$CH_3O_2$] was ~ $10^{13}$ molecule cm$^{-3}$ (Dagaut and Kurylo, 1990;Roehl et al., 1996). As the absorption cross sections of $Cl_2$ and $CH_3CHO$ at 250 nm lay in the range $10^{-21}$–$10^{-22}$ cm$^2$ molecule$^{-1}$ (Keller-Rudek et al., 2013), the unaccounted for absorption of these species may have led to an overestimation of $\sigma_{250nm}(CH_3O_2)$.

As noted in the 2001 review by Tyndall et al. (Tyndall et al., 2001), none of the previous laboratory studies of the $CH_3O_2$
recombination measured [$CH_3O_2$] by any method other than UV–spectroscopy. In addition, the traditional time–resolved measurements of $CH_3O_2$ used high $CH_3O_2$ concentrations ($10^{13}$–$10^{15}$ molecule cm$^{-3}$) and, as the self–reaction is fairly slow, Tyndall et al. stated that the results were potentially affected by secondary chemistry (Tyndall et al., 2001). Therefore, there is a need for the use of a complementary technique in the kinetic study of this reaction, for example by LIF as described in this paper, which may offer some advantages to probe $CH_3O_2$ selectively in the absence of interferences from other species. In
addition, LIF is more sensitive and hence requires significantly lower radical concentrations ([$CH_3O_2$]$_0 = (1–3) \times 10^{11}$ molecule cm$^{-3}$ here) than for the UV–absorption studies which may help to minimize potential secondary chemistry.

### 3.3 Methoxy radical measurement within HIRAC

The typical concentration of [$O_2$] = $5 \times 10^{18}$ molecule cm$^{-3}$ used in the HIRAC experiments described above was lowered in
some experiments to decrease the consumption of $CH_3O$ by $O_2$ via Reaction (R4). In this manner, a concentration of methoxy radicals was obtained above the FAGE limit of detection in HIRAC to enable a direct measurement over few minutes. The



chamber was filled with high purity nitrogen (> 99.998 %), but the ~ 6 m long $N_2$ delivery pipe was purposely incompletely purged before the experiment in order to deliver trace levels of oxygen to HIRAC. The initial $Cl_2$ concentration in these experiments was $5.6 \times 10^{15}$ molecule $cm^{-3}$ and hence is 1–2 orders of magnitude higher than $[Cl_2]_0$ used in the kinetic experiments above in order to generate higher [Cl] and hence $[CH_3O]$. The temporal profile of $CH_3O$ is shown in Fig. 8,

together with a numerical simulation of $CH_3O(t)$ using a chemistry system described in the Supplementary Information. The best fit to the experimental $CH_3O$ concentration profile was obtained for $[O_2] = (5.4 \pm 0.6) \times 10^{15}$ molecule $cm^{-3}$, i.e. around 0.02 % relative to $N_2$. The numerical simulations showed that $Cl_2$ consumption was dominated by the reaction with $CH_3$ radicals, present at a relatively high concentration, explaining the ~ 50 % decrease in $[CH_3O]$ observed during its temporal measurement shown in Fig. 8. The Supplementary Information (Fig. S5) shows the concentration profiles of $Cl_2$, Cl, $CH_3$ and

$CH_3O_2$ obtained by numerical simulations performed over ~ 2 min.

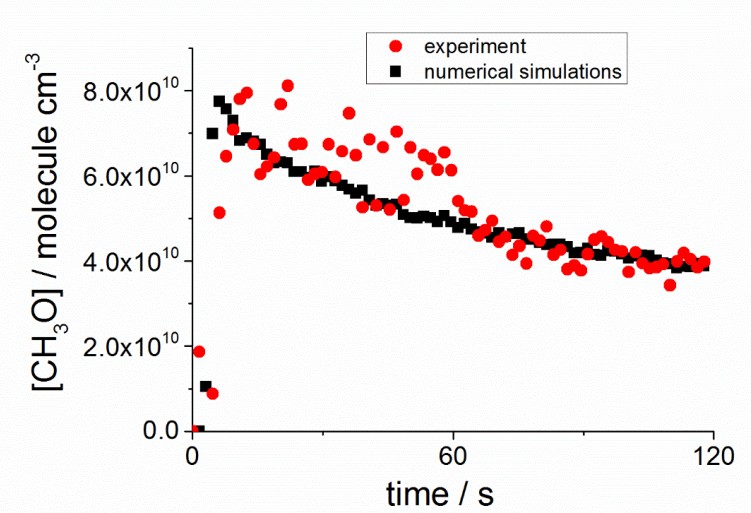

**Figure 8.** Concentration of $CH_3O$ as a function of time in HIRAC: red points are the experimental data and black points are

generated by a numerical simulation. $CH_3O$ radicals were formed as a product of the self-reaction of $CH_3O_2$ species at 295 K and 1 bar of $N_2$, with $CH_3O_2$ generated by the reaction of Cl atoms with $CH_4$, with the HIRAC black lamps being turned on at time zero. Oxygen was present in trace amounts, determined to be $(5.4 \pm 0.6) \times 10^{15}$ molecule $cm^{-3}$ from comparison of the simulations to the experimental data. The chemical mechanism used in the numerical simulations is presented in the Supplementary Information.

These results demonstrate the capability to measure an absolute concentration of $CH_3O$ radicals in a simulation chamber, with $CH_3O$ representing a further model target species for the validation of chemical mechanisms for the chemical oxidation of VOCs. However, it is recognized that the experiments need to be performed at reduced $[O_2]$, and that $[O_2]$ needs to be known *a priori* in order to test robustly the accuracy of the chemical mechanism and underlying kinetic parameters.

**4. Conclusions**

Currently there is no measurement of the absolute concentration of $CH_3O_2$ radicals in the atmosphere. In this work the FAGE technique has been extended by adding the capability to detect $CH_3O_2$ and $CH_3O$ radicals to the more typical measurement of





OH and HO$_2$ radicals. The method enables the speciated and sensitive detection of CH$_3$O$_2$ radicals by converting CH$_3$O$_2$ into CH$_3$O by reaction with NO and detecting the resultant CH$_3$O by LIF with excitation at *ca*. 298 nm. The limit of detection of the method obtained using the radical source commonly employed to provide accurate concentrations of OH with added CH$_4$, is $3.8 \times 10^8$ molecule cm$^{-3}$ for a signal-to-noise ratio of 2 and 5 min time resolution and reduces to $1.1 \times 10^8$ molecule cm$^{-3}$ for

*S/N* = 2 and 1 hour averaging time. Therefore, the method has the potential to be used in field measurements of the diurnal profiles of CH$_3$O$_2$ in clean air with low NO$_x$ levels, such as remote continental environments and in the marine boundary layer. Further improvements of the FAGE sensitivity could be achieved via the increase in the laser repetition frequency above the current value of 5 kHz, a decrease in the detection chamber pressure (currently ~ 2.65 Torr), and the use of a shorter distance between the inlet sampling pinhole and the fluorescence detection axis (presently a long distance of ~ 580 mm). The method

is also demonstrated for the direct detection of CH$_3$O, in the absence of added NO to the fluorescence cell. The limit of detection for CH$_3$O determined using the conventional radical source for *S/N* = 2 and 5 min averaging time is $3.0 \times 10^8$ molecule cm$^{-3}$.

Additional investigations into the FAGE sensitivity for CH$_3$O$_2$ were carried out in the HIRAC simulation chamber at Leeds, by studying the kinetics of the second–order decays of CH$_3$O$_2$ by its self–reaction. The second–order decays of CH$_3$O$_2$ were

analysed by fixing the observed rate coefficient to the IUPAC recommendation, $k_{obs}$ = $(4.8 \pm 1.1) \times 10^{-13}$ cm$^3$ molecule$^{-1}$ s$^{-1}$, (Atkinson et al., 2006) in the fitting routine to extract the FAGE sensitivity factor for CH$_3$O$_2$, $C_{CH3O2}$. The obtained value, $C_{CH3O2}$ = $(5.6 \pm 0.9) \times 10^{-10}$ counts cm$^3$ molecule$^{-1}$ s$^{-1}$ mW$^{-1}$, agrees well with the result found using the conventional radical source, $C_{CH3O2}$ = $(4.1 \pm 0.7) \times 10^{-10}$ counts cm$^3$ molecule$^{-1}$ s$^{-1}$ mW$^{-1}$ (uncertainties quoted to 1$\sigma$). The two values have overlapping error limits at 1$\sigma$ level.

In addition to the quantitative detection of CH$_3$O$_2$, experiments were carried out to measure CH$_3$O generated as a product by the CH$_3$O$_2$ self–reaction in HIRAC. Oxygen was present at a significantly lower concentration to reduce the consumption rate of CH$_3$O by reaction with O$_2$ in order to enable the measurement. Good agreement between the experimental data and [CH$_3$O] generated by numerical simulations using a model describing the chemical system was obtained, demonstrating the capability to quantitatively measure CH$_3$O. As well as CH$_3$O$_2$, a measurement of CH$_3$O will be useful as a further model target

in future mechanistic studies of atmospherically relevant chemical systems within HIRAC.

**Acknowledgements**

This work was supported by the Natural Environment Research Council (grant number NE/M011208/1) and the National Centre for Atmospheric Science, and AB is grateful to NERC for a studentship, awarded as part of the SPHERES Doctoral

Training Programme (NE/L002574/1). The authors thank Bethany Ronnie for help with the measurements of methyl peroxy concentrations across HIRAC.

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
