# Peer review of "A new method for atmospheric detection of the CH3O2 radical"

_Atmospheric Measurement Techniques, 2017_

## Referee Comment (RC1) · Anonymous Referee #2 · 13 Jun 2017

This manuscript reports the development and the calibration of an instrument to measure ambient concentrations of methyl peroxy radicals. This instrument is based on the well-known FAGE technique, which is currently used by several groups around the world, including the authors, to measure ambient concentrations of OH and HO2 radicals. The FAGE technique was recently extended for measuring total peroxy radicals. One of the current limitations to investigate atmospheric free radicals chemistry is the lack of techniques to perform speciated measurements of peroxy radicals. The work reported in this publication starts addressing this issue by extending the use of the FAGE technique to the measurement of methyl peroxy radicals, one of the most abundant organic peroxy radicals in the atmosphere. The modified FAGE apparatus is well described and its calibration is investigated using two different approaches, the conventional water photolysis approach used to calibrate OH and HO2 on conventional FAGE instruments - with the addition of methane in the calibration cell to convert OH into CH3O2 - and an approach based on monitoring CH3O2 in an atmospheric chamber when it decays due to its self-reaction. The two approaches are shown to be in agreement within uncertainties.

This manuscript is well structured, clear and concise, and the proposed approach for measuring methyl peroxy radicals seems promising for both field measurements and laboratory studies. I therefore recommend publication in AMT after the authors address the following comments:

1/ The sensitivity of conventional FAGE instruments is known to be dependent on the ambient water concentration due to the quenching of excited OH radicals by water molecules. This matrix effect is taken into account through the calibration of the OH sensitivity at different water-vapor concentrations. Can excited CH3O radicals also be quenched by water vapor? If so, what is the implication for ambient measurements of CH3O2?

2/ For calibration purposes, CH3O2 is generated using the water-photolysis approach by adding an excess of methane in the photolysis cell. Could the authors comment on the potential quenching of excited CH3O by methane during calibration experiments?

Minor comments:

P4 L4: "Here we report he first . . ." should read "Here we report the first . . ."

P4 L13: Please report the sampling flow rate of the FAGE apparatus

P5 L4-5: Since the detection of the CH3O fluorescence is red-shifted from the excitation, why is the counting window delayed by 100 ns from the laser pulse? This time gating approach is usually used for the detection of on-resonant fluorescence.

P5 L12-15: the authors mention that the wavelength is tuned on/off resonance with the CH3O transition line. In FAGE instruments, OH is continuously generated in a

reference cell to be able to precisely tune the laser wavelength on and off resonance. How is it performed for CH3O on this instrument? Is CH3O continuously generated in a reference cell? If so, how is it done?

P7 L15: The authors indicate a CH3O2-to-CH3O conversion efficiency of 40% at the optimum NO concentration. However, since CH3O can also be lost through its reaction with NO (and potentially through its reaction with O2 as well), isn't the 40% representative of a lower limit of the conversion?

P10 L12-13: It is indicated that the photon flux was varied between 0-1.5E14 photon/cm2/s. However, the lower bound reported for the radical generation is 1.5E10 molecule/cm3, which cannot correspond to a photon flux set at zero. Please clarify.

P12 L9-14: The detection limits are calculated for a BKG signal of approximately 100 ct/s, which is reported as a typical value for this instrument. What are the contributions of the scattered visible and laser lights? How is the BKG signal expected to change when the solar irradiation changes during field measurements? How will it affect the detection limit during daytime?

P13 L22: caption Fig. 7. "cm-1" should read "cm-3"

P15 L21 & L22: Two different uncertainties are given for the on-line signal: 12% and 6%. Which one is correct?

P16 L39: The authors indicate that the oxygen concentration was lowered in some experiments performed on the HIRAC chamber. Could the lower oxygen concentration lead to a different sensitivity towards CH3O due to changes in quenching rates of excited CH3O?

---

## Referee Comment (RC2) · Anonymous Referee #3 · 17 Aug 2017

This paper presents details on the development of a new method to detect the CH$_3$O$_2$ radical in the atmosphere using laser-induced fluorescence techniques. Current methods for the detection of peroxy radicals in the atmosphere are unable to distinguish between CH$_3$O$_2$ and other organic peroxy radicals. Given the importance of CH$_3$O$_2$ radical chemistry in the atmosphere, a selective and sensitive method to detect these radicals both in the atmosphere and in chamber studies would provide an important tool for improving our understanding of atmospheric chemistry.

The method utilizes the Laser-Induced Fluorescence-Fluorescence Assay by Gas Expansion (LIF-FAGE) technique that is currently used for the sensitive detection of the OH and HO$_2$ radicals in the atmosphere. Similar to the detection of HO$_2$ radicals by this technique, the authors convert CH$_3$O$_2$ radicals to CH$_3$O radicals inside the FAGE

detection cell using the $CH_3O_2 + NO$ reaction, and then detect the $CH_3O$ radicals using laser-induced fluorescence. The paper describes several methods used to calibrate the instrument – production of $CH_3O_2$ radicals from the $OH + CH_4$ reaction in a flow tube (with OH radicals produced from the photolysis of water vapor), production of $CH_3O_2$ from the photolysis of $CH_3OH$ in a flow tube, and monitoring the decay of $CH_3O_2$ radicals from the $CH_3O_2 + CH_3O_2$ reaction in the HIRAC chamber. The paper is well written and suitable for publication in AMT after the authors consider the following comments.

Page 4 line 30 and Page 6 Figures 2 and 3: As with detection of OH by the LIF-FAGE technique, the authors must tune the laser on and off of the $CH_3O$ transition to determine the net signal due to $CH_3O$ fluorescence and the background signal due to laser scatter and other broadband fluorescence. OH LIF-FAGE instruments use a reference cell that generates high concentration of OH radicals to ensure that the laser is tuned to the correct frequency. It is unclear how the authors know that the laser is tuned to the correct $CH_3O$ excitation wavelength. Do they use a spectrometer to measure the wavelength, or do they have a reference cell that generates $CH_3O$ radicals?

Page 8 line 25: Equation 2 assumes that the concentration of methanol is proportional to the concentration of water vapor and that any loss of methanol in their bubbler system is equal to any loss of water in their flow tube. Can the authors justify this assumption?

Page 12, line 25: The authors claim that reducing the pressure in their FAGE detection cell could increase the sensitivity of the instrument. Is this due to reduced quenching of the $CH_3O$ fluorescence by air? Have the authors measured the impact trace gases on the fluorescence efficiency, such as water vapor?

Page 12, line 26: How does the OH sensitivity of the HIRAC FAGE compare to the field instrument? Assuming the $CH_3O$ sensitivity scales with the differences in the OH sensitivity, can the authors be more specific regarding the potential improvement in the

LOD if this technique were to be used in the field instrument?

Page 16, line 19: The authors suggest that based on their flow tube calibrations that the rate constant for the $CH_3O_2$ + $CH_3O_2$ reaction may be 25% too high, perhaps due to a 25% overestimation of the $CH_3O_2$ absorption cross section. What is the uncertainty associated with the recommended rate constant? Does the rate constant derived using their flow tube calibration factor agree to within the combined uncertainty of the calibration and the rate constant?

Figure 8: The authors measure the concentration of $CH_3O$ in nitrogen to reduce the loss of $CH_3O$ from the $CH_3O_2$ + $O_2$ reaction. However, it appears that they use the calibration factor determined in air to estimate the $CH_3O$ concentrations in this experiment. Does the calibration factor change in $N_2$ compared to air due to different fluorescence quenching rates?

---

## Author Comment (AC1) · 13 Sep 2017

**Author response to anonymous referee #3 on "A new method for atmospheric detection of the CH$_3$O$_2$ radical" by L. Onel et al.**

Note: The changes in the manuscript addressing the comments of the referee #3 are highlighted in yellow below. The authors refer to the line numbers in the manuscript before revision mentioned in the comments.

The authors would like to thank anonymous referee #3 for their valuable comments to this manuscript.

***Page 4 line 30 and Page 6 Figures 2 and 3***: *As with detection of OH by the LIF FAGE technique, the authors must tune the laser on and off of the CH3O transition to determine the net signal due to CH3O fluorescence and the background signal due to laser scatter and other broadband fluorescence. OH LIF-FAGE instruments use a reference cell that generates high concentration of OH radicals to ensure that the laser is tuned to the correct frequency. It is unclear how the authors know that the laser is tuned to the correct CH3O excitation wavelength. Do they use a spectrometer to measure the wavelength, or do they have a reference cell that generates CH3O radicals?*

The signals were large enough that during conditions where CH$_3$O$_2$ concentrations were constant (e.g. in calibrations or during HIRAC experiments where steady-state concentrations were generated) it was established that the laser-wavelength was stable over a long period once the laser wavelength had been tuned to the CH$_3$O transition. Hence, the online wavelength position for CH$_3$O fluorescence detection was found without using a reference cell. The laser excitation scans shown in Figures 2 and 3 were performed using the flow tube method described in the sections 2.3.1 and 2.3.2.1 to generate either CH$_3$O (by the CH$_3$OH photolysis at 185 nm) or CH$_3$O$_2$ (by the H$_2$O photolysis at 185 nm to generate OH followed by the reaction of the produced OH with CH$_4$ in the presence of O$_2$).

In the HIRAC experiments the concentration of CH$_3$O$_2$ radicals generated in the chamber in a steady-state with the UV lamps turned on at the beginning of each experiment using the Cl$_2$/CH$_4$/air system was used to tune the laser at the correct excitation wavelength by performing similar scans to the laser scans shown in Figure 3.

In all measurements the offline wavelength position was fixed to the value obtained by adding 2.5 nm to λ(online) as described at page 5, line 14. For field measurements in the future, when the concentrations of CH$_3$O$_2$ (and hence CH$_3$O after conversion) will be both lower and more variable over short timescales, a reference cell will be necessary. We are in the process of developing a reference cell.

The third paragraph on page 5 (lines 10-24) was changed for clarification:

"The signals were large enough that during conditions where $CH_3O_2$ concentrations were constant (e.g. in calibrations or during HIRAC experiments where steady-state concentrations were generated) it was established that the laser-wavelength was stable over a long period once the laser wavelength had been tuned to the $CH_3O$ transition. Hence, the online wavelength position for $CH_3O$ fluorescence detection was found without using a reference cell. Figure 2 shows the laser excitation spectrum centred at ~298 nm in the $v_3$ vibronic band recorded using an increment of $\Delta\lambda = 10^{-3}$ nm. The spectrum agrees well with previous work (Inoue et al., 1980;Kappert and Temps, 1989;Shannon et al., 2013). Figure 3 shows typical laser excitation scans performed over a narrower range of wavelengths in order to locate $\lambda$(online). The LIF spectra were obtained by using the $CH_3O$ or $CH_3O_2$ radicals generated in a flow tube described in Sect. 2.3.1, with the flow tube output impinged close to the FAGE sampling inlet. The radicals were generated using the 184.9 nm light output of a Hg Pen-Ray lamp by either the photolysis of methanol in nitrogen to generate $CH_3O$ or the photolysis of water vapour in synthetic air (to generate OH) in the presence of methane to form $CH_3O_2$. The $CH_3O$ radicals were directly detected, while the $CH_3O_2$ radicals were first converted to $CH_3O$ species by added NO prior to the fluorescence detection cell (Fig. 1). Similar laser scans to the scans shown in Fig. 3 were recorded by using the $CH_3O_2$ radicals produced in a steady-state concentration in HIRAC using photolytic mixtures of $Cl_2/CH_4$/air as described in Sect. 2.3.2.2. There were no unexpected features in the laser scans recorded when FAGE sampled $CH_3O_2$ radicals from HIRAC, consistent with no interference being anticipated in the FAGE measurements of $CH_3O$ as there were no other species in HIRAC absorbing at 298 nm and fluorescing at the wavelengths transmitted by the bandpass filter (average transmission > 80 % over 320 – 430 nm).

In this work the FAGE signals were large enough that during conditions where $CH_3O_2$ concentrations were constant (e.g. in calibrations or during HIRAC experiments where steady-state concentrations were generated) it was established that the laser wavelength was stable over a long period once $\lambda$ had been tuned to the $CH_3O$ transition. Hence, $\lambda$(online) was found without using a reference cell. We are in the process of developing a reference cell for field measurements in the future, when the concentrations of $CH_3O_2$ (and hence $CH_3O$ after conversion) will be both lower and more variable over short timescales."

*Page 8 line 25: Equation 2 assumes that the concentration of methanol is proportional to the concentration of water vapor and that any loss of methanol in their bubbler system is equal to any loss of water in their flow tube. Can the authors justify this assumption?*

Equation 2 assumes that the concentration of methanol vapour in the photolysis flow tube is equal to the concentration of water vapour in the flow tube obtained when the bubbler contained water instead of methanol. The flow tube calibration using the water vapour photolysis represents the conventional FAGE calibration method for OH and $HO_2$ and previous investigations have shown that the water vapour loss in the system formed by the bubbler and the flow tube is negligible. Even less wall losses can be expected in the case of methanol, which has a significantly higher vapour pressure than water.

The following sentence was added after equation 2 (page 8, line 29):

"Equation 2 assumes that there were no losses of water vapour and methanol vapour by condensation in the tubing connecting the bubbler to the flow tube. This is as expected based on the small difference in temperature between the bubbler (*vide supra*) and the connecting tubing (typically held at ~ 20 $^\circ$C) and as the gas going through the bubbler was diluted with the gas by-passing the bubbler."

***Page 12, line 25***: *(i) The authors claim that reducing the pressure in their FAGE detection cell could increase the sensitivity of the instrument. Is this due to reduced quenching of the CH3O fluorescence by air? (ii) Have the authors measured the impact trace gases on the fluorescence efficiency, such as water vapor?*

**(i)** A potential improvement of the instrument sensitivity for $CH_3O_2$ by using a pressure in the detection cell lower than the present limit of 2.65 Torr is expected because of the experimental observation of an increase in the fluorescence signal when the pressure in the detection cell is reduced from 10.00–2.65 Torr. As the pressure is reduced there is a reduction in the $CH_3O$ number density (which would decrease the LIF signal) and also a decrease in the quenching rate of the $CH_3O$ fluorescence by air, and hence an increase in the fluorescence quantum yield (which would increase the LIF signal). These two effects are opposing, but at low pressures do not cancel, leading to the observed increase in signal with lower pressures. It is therefore expected that as the pressure is reduced further below 2.65 Torr that the signal would continue to increase. Another reason could be that the characteristics of the jet expansion and/or the ensuing flow to the LIF detection region change with pressure, leading to a more favourable transmission of radicals to the detection region, but it is difficult to test this experimentally. For clarification the text (page 12, lines 24 – 27) was modified as follows:

"The present investigations into the change of sensitivity with pressure in the range from 2.65–10.00 Torr found that 2.65 Torr is the optimum value in this pressure interval. The result suggests that, by reducing the pressure in the above range of values, the decrease in fluorescence due to the reduction in the $CH_3O$ number density was overcome by the increase in the fluorescence quantum yield due to a lower fluorescence quenching rate. Another reason could be that the characteristics of the jet expansion and/or the ensuing flow to the LIF detection region change with pressure, leading to a more favourable transmission of radicals to the detection region, but it is difficult to test this experimentally. Hence an additional improvement in the sensitivity might be obtained by using a lower detection cell pressure than the current value of 2.65 Torr using a more powerful pump."

**(ii)** No measurement of the rate coefficients of the fluorescence quenching by the traces gases has been performed in this work. However, a very good agreement was obtained between the flow tube calibrations for $CH_3O_2$ with two different concentrations of water vapour in the flow tube: $7.5 \times 10^{16}$ molecule cm$^{-3}$ or $3 \times 10^{17}$ molecule cm$^{-3}$ (corresponding to $2.6 \times 10^{14}$

molecule cm$^{-3}$ and 1.0 x $10^{15}$ molecule cm$^{-3}$, respectively in the FAGE detection cell) as shown by Figure 6 in Sect. 2.3.2.1. The result presented in Figure 6 shows that the $CH_3O$ fluorescence quenching rate by water is minor for the above $[H_2O]$.

Methane was also present in the FAGE chamber in concentrations of several times $10^{14}$ molecule cm$^{-3}$. Calculations using the $CH_3O$ fluorescence quenching rate coefficient of $CH_4$ reported by Wantuck et al. (1987), $1.05 \times 10^{-10}$ s$^{-1}$, and a pressure in the FAGE detection cell of 2.65 Torr show only minor decreases in the fluorescence quantum yield, by few percent, when $[CH_4]$ is increased from zero to the experimental values. Assuming a quenching rate coefficient of $H_2O$ equal to that of $CH_4$, similar small decreases in the fluorescence quantum yield were computed when $[H_2O]$ was increased from zero to the concentration values used in the flow tube calibration (0.3 - 1.0 x $10^{15}$ molecule cm$^{-3}$).

A paragraph which discusses the $CH_3O(A)$ quenching rates of water and methane at the concentrations used in the flow tube calibration of the FAGE instrument for $CH_3O_2$ has been added at the end of the section 3.1.1:

"The calibrations using the flow-tube ("wand") method have been performed under water vapour concentrations similar to the ambient $[H_2O_{vapour}]$ but few orders of magnitude higher than those present in the HIRAC chamber experiments. In contrast with $[H_2O_{vapour}]$ the methane concentrations used in the "wand" method were similar to $[CH_4]$ present in HIRAC but higher than $[CH_4]$ in the atmosphere. However, as detailed in this paragraph, the effects of methane and water on our sensitivity are minimal. Estimations using the reported fluorescence quenching rate coefficient of $CH_3O(A)$ by $CH_4$, $k_{quench.CH4} = 1.05 \times 10^{-10}$ s$^{-1}$, (Wantuck et al., 1987) and the concentrations of $CH_4$ in the LIF detection cell for the calibrations using the flow-tube ($1.7 \times 10^{14}$ molecule cm$^{-3}$ and $3.4 \times 10^{14}$ molecule cm$^{-3}$, corresponding to $5.0 \times 10^{16}$ molecule cm$^{-3}$ and $1.0 \times 10^{17}$ molecule cm$^{-3}$, respectively in the flow tube) resulted in only ~ 1–2% lower fluorescence quantum yield compared to the value determined in the absence of $CH_4$. No literature value has been found for the fluorescence rate coefficient of $CH_3O(A)$ fluorescence by $H_2O$ vapour. However, even if it assumed to be as large as the above reported value for $CH_4$ ($k_{quench.CH4}$), only a few percent decrease in the fluorescence quantum yield is computed (compared with a water concentration of zero) for the levels of $H_2O$ vapour which are present at the $CH_3O_2$ FAGE detection axis when using the flow tube calibration method. These levels (1–2 % v/v) are similar to a typical water vapour concentration in the atmosphere. A very good agreement has been obtained between the calibration factors for $CH_3O_2$ detection with two different concentrations of water vapour in the flow tube: $7.5 \times 10^{16}$ molecule cm$^{-3}$ or $3.0 \times 10^{17}$ molecule cm$^{-3}$ (corresponding to 2.6 $\times 10^{14}$ molecule cm$^{-3}$ and $1.0 \times 10^{15}$ molecule cm$^{-3}$, respectively in the FAGE cell) as shown in Figure 6 in Sect. 2.3.2.1. This very good agreement for $H_2O$ vapour and the above calculations for $CH_4$ support the use of the flow tube method for the FAGE calibration of the $CH_3O_2$ concentrations."

***Page 12, line 26***: *How does the OH sensitivity of the HIRAC FAGE compare to the field instrument? Assuming the CH3O sensitivity scales with the differences in the OH sensitivity, can the authors be more specific regarding the potential improvement in the LOD if this technique were to be used in the field instrument?*

The HIRAC FAGE sensitivity for OH is about two times lower than the ground-based field instrument sensitivity for OH: $C_{OH \text{ (HIRAC)}} = 8 \times 10^{-8}$ counts cm$^3$ molecule$^{-1}$ s$^{-1}$ mW$^{-1}$, $C_{OH \text{ (field)}} = 1.5 \times 10^{-7}$ counts cm$^3$ molecule$^{-1}$ s$^{-1}$ mW$^{-1}$.

As the distance from the inlet pinhole to the laser axis in the $CH_3O_2$ fluorescence cell (Figure 1, 580 mm) is considerably longer than the corresponding distance in the ground–based field fluorescence cell for OH and $HO_2$ detection (88 mm), improvements in the $CH_3O_2$ sensitivity are expected for the field FAGE instrument. The decrease in the pinhole–to–laser axis from 580 mm to 88 mm would result in a reduced loss of the $CH_3O_2$ radicals on the instrument internal walls and would provide a greater population in the laser probed rotational level as the gas is still cooler than ambient following the pinhole expansion. However, the increase in the $CH_3O$ sensitivity cannot be quantified simply using the difference in the OH sensitivity between the HIRAC instrument and the field instrument. A larger increase in sensitivity between the field instrument and HIRAC would be expected for OH than for $CH_3O_2$ based on expected heterogeneous losses, as the wall loss of OH is larger than the wall loss of $CH_3O_2$. However, how much the decrease in temperature at the laser axis owing to a smaller nozzle-to-laser axis distance improves the FAGE sensitivity for $CH_3O$ compared to the sensitivity for OH needs further investigation.

We think that no modification of the text is necessary as it cannot be assumed that improvements in the $CH_3O_2$ sensitivity will scale with the difference in the OH sensitivity between the HIRAC instrument and the ground-field instrument.

***Page 16, line 19***: *The authors suggest that based on their flow tube calibrations that the rate constant for the CH3O2 + CH3O2 reaction may be 25% too high, perhaps due to a 25% overestimation of the CH3O2 absorption cross section. What is the uncertainty associated with the recommended rate constant? Does the rate constant derived using their flow tube calibration factor agree to within the combined uncertainty of the calibration and the rate constant?*

The associated uncertainty with the IUPAC recommended value of the rate coefficient for the $CH_3O_2$ self-reaction, $k_{CH3O2}$, is ~ 12% (1$\sigma$). Our measured value, based on the flow tube calibration factor is ~25% lower than the IUPAC recommendation, with an overall error of ~20% (1$\sigma$). Therefore, the obtained $k_{CH3O2}$ have overlapping error limits with the IUPAC preferred value at the 1$\sigma$ level.
The overall uncertainties of the two calibration methods of FAGE are discussed in detail in the manuscript. Even though the kinetic method agrees well with the flow tube method, it should be noted that the use of a lower value of $k$ than $k_{CH3O2}$(IUPAC) would improve the level of agreement. Therefore, the text has not been changed.

***Page 17, Figure 8****: The authors measure the concentration of CH3O in nitrogen to reduce the loss of CH3O from the CH3O2 + O2 reaction. However, it appears that they use the calibration factor determined in air to estimate the CH3O concentrations in this experiment. Does the calibration factor change in N2 compared to air due to different fluorescence quenching rates?*

The concentration of $CH_3O$ in the HIRAC experiment shown in Figure 8 was obtained by using the calibration factor for methoxy radicals, which in turn was determined using the photolysis of methanol in $N_2$ method described in section 2.3.1. In this HIRAC experiment $O_2$ was only present in trace amounts ($[O_2]_{HIRAC}$ = 5.4 × $10^{15}$ molecule $cm^{-3}$, which corresponds to 1.8 x $10^{13}$ molecule $cm^{-3}$ $O_2$ in the fluorescence detection cell) as described in section 3.3. This $[O_2]$ is too small to produce a faster quenching rate of the $CH_3O$ LIF signal in the chamber experiment compared to the quenching rate when using pure $N_2$, as estimated using the quenching rate coefficient of $O_2$ reported by Wantuck et al. (1986), 2.5 × $10^{-11}$ $cm^3$ $molecule^{-1}$ $s^{-1}$.

The following sentence was added in the first paragraph of section 3.3 for clarification:

"The concentration of $CH_3O$ during the experiment was computed by using the FAGE calibration factor for methoxy radicals generated from the photolysis of methanol in $N_2$, $C_{CH3O}$ = (5.1 ± 2.2) ×$10^{-10}$ counts $cm^3$ $molecule^{-1}$ $s^{-1}$ $mW^{-1}$ (Sect. 3.1.1). The temporal profile of the $CH_3O$ is shown in Fig. 8…"

---

## Author Comment (AC2) · 13 Sep 2017

**Author response to anonymous referee #2 on "A new method for atmospheric detection of the CH$_3$O$_2$ radical" by L. Onel et al.**

Note: The changes in the manuscript addressing the comments of the referee #2 are highlighted in yellow below. The authors refer to the line numbers in the manuscript before revision mentioned in the comments.

The authors would like to thank anonymous referee #2 for their valuable comments to this manuscript.

The first two questions (1/ and 2/) address the quenching of the CH$_3$O fluorescence by water vapour and methane, respectively:

*1/ The sensitivity of conventional FAGE instruments is known to be dependent on the ambient water concentration due to the quenching of excited OH radicals by water molecules. This matrix effect is taken into account through the calibration of the OH sensitivity at different water-vapor concentrations. Can excited CH3O radicals also be quenched by water vapor? If so, what is the implication for ambient measurements of CH3O2?*

*2/ For calibration purposes, CH3O2 is generated using the water-photolysis approach by adding an excess of methane in the photolysis cell. Could the authors comment on the potential quenching of excited CH3O by methane during calibration experiments?*

No measurement of the rate coefficients of the fluorescence quenching by the traces gases has been performed in this work. However, a very good agreement was obtained between the flow tube calibrations for CH$_3$O$_2$ with two different concentrations of water vapour in the flow tube: 7.5 x 10$^{16}$ molecule cm$^{-3}$ or 3 x 10$^{17}$ molecule cm$^{-3}$ (corresponding to 2.6 x 10$^{14}$ molecule cm$^{-3}$ and 1.0 x 10$^{15}$ molecule cm$^{-3}$, respectively in the FAGE detection cell) as shown by Figure 6 in Sect. 2.3.2.1. The result presented in Figure 6 shows that the CH$_3$O fluorescence quenching rate by water is minor for the above [H$_2$O].

Methane was also present in the FAGE chamber in concentrations of several times 10$^{14}$ molecule cm$^{-3}$. Calculations using the CH$_3$O fluorescence quenching rate coefficient of CH$_4$ reported by Wantuck et al. (1987), $1.05 \times 10^{-10}$ s$^{-1}$, and a pressure in the FAGE detection cell of 2.65 Torr show only minor decreases in the fluorescence quantum yield, by few percent, when [CH$_4$] is increased from zero to the experimental values. Assuming a quenching rate coefficient of H$_2$O equal to that of CH$_4$, similar small decreases in the fluorescence quantum yield were computed when [H$_2$O] was increased from zero to the concentration values used in the flow tube calibration (0.3 - 1.0 x 10$^{15}$ molecule cm$^{-3}$). Therefore, the effects of methane and water on the FAGE sensitivity for CH$_3$O$_2$ are minimal.

A paragraph which discusses the CH$_3$O(*A*) quenching rates of water and methane at the concentrations used in the flow tube calibration of the FAGE instrument for CH$_3$O$_2$ has been added at the end of the section 3.1.1:

"The calibrations using the flow-tube ("wand") method have been performed under water vapour concentrations similar to the ambient [$H_2O_{vapour}$] but few orders of magnitude higher than those present in the HIRAC chamber experiments. In contrast with [$H_2O_{vapour}$] the methane concentrations used in the "wand" method were similar to [$CH_4$] present in HIRAC but higher than [$CH_4$] in the atmosphere. However, as detailed in this paragraph, the effects of methane and water on our sensitivity are minimal. Estimations using the reported fluorescence quenching rate coefficient of $CH_3O(A)$ by $CH_4$, $k_{quench.CH4} = 1.05 \times 10^{-10}$ s$^{-1}$, (Wantuck et al., 1987) and the concentrations of $CH_4$ in the LIF detection cell for the calibrations using the flow-tube ($1.7 \times 10^{14}$ molecule cm$^{-3}$ and $3.4 \times 10^{14}$ molecule cm$^{-3}$, corresponding to $5.0 \times 10^{16}$ molecule cm$^{-3}$ and $1.0 \times 10^{17}$ molecule cm$^{-3}$, respectively in the flow tube) resulted in only ~ 1–2% lower fluorescence quantum yield compared to the value determined in the absence of $CH_4$. No literature value has been found for the fluorescence rate coefficient of $CH_3O(A)$ fluorescence by $H_2O$ vapour. However, even if it assumed to be as large as the above reported value for $CH_4$ ($k_{quench.CH4}$), only a few percent decrease in the fluorescence quantum yield is computed (compared with a water concentration of zero) for the levels of $H_2O$ vapour which are present at the $CH_3O_2$ FAGE detection axis when using the flow tube calibration method. These levels (1–2% v/v) are similar to a typical water vapour concentration in the atmosphere. A very good agreement has been obtained between the calibration factors for $CH_3O_2$ detection with two different concentrations of water vapour in the flow tube: $7.5 \times 10^{16}$ molecule cm$^{-3}$ or $3.0 \times 10^{17}$ molecule cm$^{-3}$ (corresponding to 2.6 $\times 10^{14}$ molecule cm$^{-3}$ and $1.0 \times 10^{15}$ molecule cm$^{-3}$, respectively in the FAGE cell) as shown in Figure 6 in Sect. 2.3.2.1. This very good agreement for $H_2O$ vapour and the above calculations for $CH_4$ support the use of the flow tube method for the FAGE calibration of the $CH_3O_2$ concentrations."

**Minor comments**

*P4 L4: "Here we report he first ..." should read "Here we report the first ..."*

The suggested correction has been made.

*P4 L13: Please report the sampling flow rate of the FAGE apparatus*

Now the sampling flow rate is given at the beginning of section 2.1 (page 4):
"The gas was sampled with a flow rate of 3.2 slm through a 1 mm diameter pinhole…"

*P5 L4-5: Since the detection of the CH3O fluorescence is red-shifted from the excitation, why is the counting window delayed by 100 ns from the laser pulse? This time gating approach is usually used for the detection of on-resonant fluorescence.*

The off-resonance $CH_3O$ fluorescence occurs between ~ 300 – 400 nm and, hence a relatively broad bandpass filter, with an average transmission > 80% between 320–430 nm, was used for the fluorescence collection. However, it appears that red-shifted scattered laser light (the excitation wavelength was ~ 298 nm) produced in the FAGE chamber also passed through

the interference filter, increasing the background. In order to avoid the majority of these background counts, the gate unit was opened 100 ns after the probe light pulse. As the optimum gate-width found for the $CH_3O$ fluorescence was 2 µs (*vide infra*), no significant loss of $CH_3O$ signal was encountered by the 100 ns delay in the fluorescence detection. Future improvements to the instrument will improve changing the cell material or coating to reduce this scattered light background.

The second paragraph on page 5 was changed as follows:

"The relatively broad bandpass filter used for the collection of the $CH_3O$ fluorescence (average transmission > 80% between 320–430 nm) allowed some red-shifted scattered light (presumably from the walls of the chamber) generated by the probe laser to be transmitted and hence detected by the MCP-PMT. In order to ameliorate this and reduce the background signal, the gate unit was opened 100 ns after the laser pulse to detect fluorescence integrated over a gate-width of 2 µs. The optimum gate-width of 2 µs (values in the range 1-3 µs were compared) is consistent with the $CH_3O$ fluorescence lifetimes, calculated to be in the range of 0.9 – 1.5 µs, using the reported radiative lifetimes for $CH_3O$ of 1.5 µs (Inoue et al., 1979), 2.2 µs (Ebata et al., 1982) and (4 ± 2) µs (Wendt and Hunziker, 1979) and using the fluorescence quenching rate coefficients of $N_2$ and $O_2$ (Wantuck et al., 1987) to calculate the rate of quenching at the pressure in the FAGE detection cell ((2.65 ± 0.05) Torr). As the fluorescence lifetime of $CH_3O(A)$ in the detection cell is 0.9–1.5 µs, delaying the counting of the fluorescence by 100 ns makes very little difference (88–91%) in the fraction of fluorescence collected."

*P5 L12-15: The authors mention that the wavelength is tuned on/off resonance with the CH3O transition line. In FAGE instruments, OH is continuously generated in a reference cell to be able to precisely tune the laser wavelength on and off resonance. How is it performed for CH3O on this instrument? Is CH3O continuously generated in a reference cell? If so, how is it done?*

The signals were large enough that during conditions where $CH_3O_2$ concentrations were constant (e.g. in calibrations or during HIRAC experiments where steady-state concentrations were generated) it was established that the laser-wavelength was stable over a long period once the laser wavelength had been tuned to the $CH_3O$ transition. Hence, the online wavelength position for $CH_3O$ fluorescence detection was found without using a reference cell. The laser excitation scans shown in Figures 2 and 3 were performed using the flow tube method described in the sections 2.3.1 and 2.3.2.1 to generate either $CH_3O$ (by the $CH_3OH$ photolysis at 185 nm) or $CH_3O_2$ (by the $H_2O$ photolysis at 185 nm to generate OH followed by the reaction of the produced OH with $CH_4$ in the presence of $O_2$).

In the HIRAC experiments the concentration of $CH_3O_2$ radicals generated in the chamber in a steady-state with the UV lamps turned on at the beginning of each experiment using the $Cl_2/CH_4$/air system was used to tune the laser at the correct excitation wavelength by performing similar scans to the laser scans shown in Figure 3.

In all measurements the offline wavelength position was fixed to the value obtained by adding 2.5 nm to $\lambda$(online) as described in the third paragraph on page 5. For field measurements in the future, when the concentrations of $CH_3O_2$ (and hence $CH_3O$ after conversion) will be both lower and more variable over short timescales, a reference cell will be necessary. We are in the process of developing a reference cell.

The third paragraph on page 5 was changed to clarify how the laser is tuned to the correct $CH_3O$ excitation wavelength:

"…Figure 2 shows the laser excitation spectrum centred at ~298 nm in the $\nu_3$ vibronic band recorded using an increment of $\Delta\lambda = 10^{-3}$ nm. The spectrum agrees well with previous work (Inoue et al., 1980;Kappert and Temps, 1989;Shannon et al., 2013). Figure 3 shows typical laser excitation scans performed over a narrower range of wavelengths in order to locate $\lambda$(online). The LIF spectra were obtained by using the $CH_3O$ or $CH_3O_2$ radicals generated in a flow tube described in Sect. 2.3.1, with the flow tube output impinged close to the FAGE sampling inlet. The radicals were generated using the 184.9 nm light output of a Hg Pen-Ray lamp by either the photolysis of methanol in nitrogen to generate $CH_3O$ or the photolysis of water vapour in synthetic air (to generate OH) in the presence of methane to form $CH_3O_2$. The $CH_3O$ radicals were directly detected, while the $CH_3O_2$ radicals were first converted to $CH_3O$ species by added NO prior to the fluorescence detection cell (Fig. 1). Similar laser scans to the scans shown in Fig. 3 were recorded by using the $CH_3O_2$ radicals produced in a steady-state concentration in HIRAC using photolytic mixtures of $Cl_2/CH_4$/air as described in Sect. 2.3.2.2. There were no unexpected features in the laser excitation scans for $CH_3O$ recorded when FAGE sampled $CH_3O_2$ radicals from HIRAC, consistent with no interference being anticipated in the FAGE measurements of $CH_3O$ as there were no other species in HIRAC absorbing at 298 nm and fluorescing at the wavelengths transmitted by the bandpass filter (average transmission > 80 % over $320 - 430$ nm). In this work the FAGE signals were large enough that during conditions where $CH_3O_2$ concentrations were constant (e.g. in calibrations or during HIRAC experiments where steady-state concentrations were generated) it was established that the laser wavelength was stable over a long period once $\lambda$ had been tuned to the $CH_3O$ transition. Hence, $\lambda$(online) was found without using a reference cell. We are in the process of developing a reference cell for field measurements in the future, when the concentrations of $CH_3O_2$ (and hence $CH_3O$ after conversion) will be both lower and more variable over short timescales."

In addition, the future construction of the reference cell is mentioned in the paragraph of section 3.1.1 where all the future instrument improvements are listed:

"The further optimizations of sensitivity and the planned construction of a reference cell to find the online wavelength position could potentially enable $CH_3O_2$ measurements to be made in urban environments where $CH_3O_2$ concentrations are estimated to be considerably lower, for example a few $10^7$ molecule $cm^{-3}$ based on modeling results (Whalley et al., to be submitted)."

*P7 L15: The authors indicate a CH3O2-to-CH3O conversion efficiency of 40% at the optimum NO concentration. However, since CH3O can also be lost through its reaction with NO (and potentially through its reaction with O2 as well), isn't the 40% representative of a lower limit of the conversion?*

The 40% value represents the optimum $CH_3O_2$ to $CH_3O$ conversion efficiency as $CH_3O$ is rapidly formed (by the $CH_3O_2$ + NO reaction) and removed in the system (by the $CH_3O$ reactions with NO and $O_2$), as discussed in section 2.2 (page 7). The text in section 2.2 explains that this result was obtained by comparison of the FAGE signal vs. [NO] generated by numerical simulations using a chemistry system formed by the above reactions with experimental data. Therefore, no text change has been made as the value of 40% was obtained from a simulation at the relevant conditions.

*P10 L12-13: It is indicated that the photon flux was varied between 0-1.5E14 photon/cm2/s. However, the lower bound reported for the radical generation is 1.5E10 molecule/cm3, which cannot correspond to a photon flux set at zero. Please clarify.*

The lower limit of the photon flux was corrected:

"The concentration of $CH_3O_2$ was varied by changing the photon flux in the range of 0.5–1.5 $\times 10^{14}$ photon cm$^{-2}$ s$^{-1}$ to generate [$CH_3O_2$] = 1.5–4.5 $\times 10^{10}$ molecule cm$^{-3}$."

*P12 L9-14: The detection limits are calculated for a BKG signal of approximately 100 ct/s, which is reported as a typical value for this instrument. What are the contributions of the scattered visible and laser lights? How is the BKG signal expected to change when the solar irradiation changes during field measurements? How will it affect the detection limit during daytime?*

The contributions to be background are roughly 50% laser scattered light within the detection cell and 50% visible light which enters the pinhole. For field measurements, there will be a contribution from solar scattered light which will scale with sunlight intensity. As for measurements of OH, the detection limit depends on the standard deviation of the background signal, and for more intense solar radiation, this will increase, increasing the detection limit. The visible scattered light is recorded on its own (together with dark counts) in a separate photon collection integration gate which is delayed a long time after the laser pulse, and is subtracted from the counts from the integration gate containing the fluorescence (after scaling for any differences in the two gate widths).

The second paragraph on page 12 was modified:

"...$BKG$ is the background signal and had a typical value of ~100 counts s$^{-1}$, which represents ~50 counts s$^{-1}$ laser scattered light within the detection cell and ~50 counts s$^{-1}$ scattered visible light which enters the pinhole from the room with a negligible contribution (1 count s$^{-1}$ on average) of the detector dark counts, $t$ is the time per data point, $m$ represents the number of online data points and $n$ is the number of offline data points."

***P13 L22***: *caption Fig. 7. "cm-1" should read "cm-3"*

"cm$^{-1}$" was changed into "cm$^{-3}$"

***P15 L21 & L22***: *Two different uncertainties are given for the on-line signal: 12% and 6%. Which one is correct?*

The paragraph on page 15 discusses the different components of the total uncertainty: 12% uncertainty represents the $2\sigma$ error in the fluorescence signal due to the uncertainty in the online wavelength position, while the 6% uncertainty is the $2\sigma$ error of the laser power measured with the power meter. A minor change was made in the last sentence of section 3.2.1:

"…of 12 % in the online FAGE signal and 6 % uncertainty in the laser power measured by the laser power meter and used to normalize the data. The uncertainty associated with the online signal, 12 % at $2\sigma$ level, was calculated as the average deviation of the signal value due to the error limits of $\pm 5 \times 10^{-4}$ nm in the online wavelength position (see the typical laser excitation scans shown in Fig. 3)."

***P16 L39***: *The authors indicate that the oxygen concentration was lowered in some experiments performed on the HIRAC chamber. Could the lower oxygen concentration lead to a different sensitivity towards CH3O due to changes in quenching rates of excited CH3O?*

Line 39 of page 16 describes the methoxy radical measurement in HIRAC (section 3.3) which is shown in Figure 8. The concentration of $CH_3O$ in these HIRAC experiments was obtained by using the calibration factor for methoxy radicals, which in turn was determined using the photolysis of methanol in $N_2$ method described in section 2.3.1. In the HIRAC experiment shown in Figure 8 $O_2$ was only present in trace amounts ($[O_2]_{HIRAC} = 5.4 \times 10^{15}$ molecule cm$^{-3}$, which corresponds to $1.8 \times 10^{13}$ molecule cm$^{-3}$ $O_2$ in the fluorescence detection cell) as described in section 3.3. This $[O_2]$ is too small to produce a faster quenching rate of the $CH_3O$ LIF signal in the chamber experiment compared to the quenching rate when using pure $N_2$, as estimated using the quenching rate coefficient of $O_2$ reported by Wantuck et al. (1986), $2.5 \times 10^{-11}$ cm$^3$ molecule$^{-1}$ s$^{-1}$.

The following sentence was added in the first paragraph of section 3.3 for clarification:

"The concentration of $CH_3O$ during the experiment was computed by using the FAGE calibration factor for methoxy radicals generated from the photolysis of methanol in $N_2$, $C_{CH3O} = (5.1 \pm 2.2) \times 10^{-10}$ counts cm$^3$ molecule$^{-1}$ s$^{-1}$ mW$^{-1}$ (Sect. 3.1.1). The temporal profile of the $CH_3O$ is shown in Fig. 8…"

---

## Author Response (AR1)

**Author response to anonymous referee #2 on "A new method for atmospheric detection of the CH3O2 radical" by L. Onel et al.**

Note: The changes in the manuscript addressing the comments of the referee #2 are highlighted in yellow below. The authors refer to the line numbers in the manuscript before revision mentioned in the comments.

The authors would like to thank anonymous referee #2 for their valuable comments to this manuscript.

The first two questions (1/ and 2/) address the quenching of the  $CH_3O$  fluorescence by water vapour and methane, respectively:

1/ The sensitivity of conventional FAGE instruments is known to be dependent on the ambient water concentration due to the quenching of excited OH radicals by water molecules. This matrix effect is taken into account through the calibration of the OH sensitivity at different water-vapor concentrations. Can excited CH3O radicals also be quenched by water vapor? If so, what is the implication for ambient measurements of CH3O2?

2/ For calibration purposes, CH3O2 is generated using the water-photolysis approach by adding an excess of methane in the photolysis cell. Could the authors comment on the potential quenching of excited CH3O by methane during calibration experiments?

No measurement of the rate coefficients of the fluorescence quenching by the traces gases has been performed in this work. However, a very good agreement was obtained between the flow tube calibrations for  $CH_3O_2$ with two different concentrations of water vapour in the flow tube: 7.5 x  $10^{16}$  molecule cm-3 or 3 x  $10^{17}$ molecule cm-3 (corresponding to 2.6 x  $10^{14}$  molecule cm-3 and 1.0 x  $10^{15}$  molecule cm-3, respectively in the FAGE detection cell) as shown by Figure 6 in Sect. 2.3.2.1. The result presented in Figure 6 shows that the CH3O fluorescence quenching rate by water is minor for the above [H2O].

Methane was also present in the FAGE chamber in concentrations of several times  $10^{14}$  molecule cm-3. Calculations using the CH3O fluorescence quenching rate coefficient of CH4 reported by Wantuck et al. (1987),  $1.05 \times 10^{-10}$  s-1, and a pressure in the FAGE detection cell of 2.65 Torr show only minor decreases in the fluorescence quantum yield, by few percent, when [CH4] is increased from zero to the experimental values. Assuming a quenching rate coefficient of H2O equal to that of CH4, similar small decreases in the fluorescence quantum yield when [H2O] was increased from zero to the concentration values used in the

flow tube calibration (0.3 - 1.0 x  $10^{15}$  molecule cm-3). Therefore, the effects of methane and water on the FAGE sensitivity for CH3O2 are minimal.

A paragraph which discusses the  $CH_3O(A)$  quenching rates of water and methane at the concentrations used in the flow tube calibration of the FAGE instrument for  $CH_3O_2$  has been added at the end of the section 3.1.1:

"The calibrations using the flow-tube ("wand") method have been performed under water vapour concentrations similar to the ambient [H2Ovapour] but few orders of magnitude higher than those present in the HIRAC chamber experiments. In contrast with  $[H_2O_{vapour}]$  the methane concentrations used in the "wand" method were similar to [CH4] present in HIRAC but higher than [CH4] in the atmosphere. However, as detailed in this paragraph, the effects of methane and water on our sensitivity are minimal. Estimations using the reported fluorescence quenching rate coefficient of CH3O(A) by CH4,  $k_{\text{quench.CH4}} = 1.05 \times 10^{-10} \text{ s}^{-1}$ , (Wantuck et al., 1987) and the concentrations of CH4 in the LIF detection cell for the calibrations using the flow-tube  $(1.7 \times 10^{14} \text{ molecule cm}^{-3} \text{ and } 3.4 \times 10^{14} \text{ molecule cm}^{-3}$ , corresponding to  $5.0 \times 10^{16} \text{ molecule cm}^{-3}$ 3 and  $1.0 \times 10^{17}$  molecule cm-3, respectively in the flow tube) resulted in only ~ 1–2% lower fluorescence quantum yield compared to the value determined in the absence of CH4. No literature value has been found for the fluorescence rate coefficient of CH3O(A) fluorescence by H2O vapour. However, even if it assumed to be as large as the above reported value for CH4 ( $k_{quench,CH4}$ ), only a few percent decrease in the fluorescence quantum yield is computed (compared with a water concentration of zero) for the levels of H2O vapour which are present at the  $CH_3O_2$  FAGE detection axis when using the flow tube calibration method. These levels (1– 2% v/v) are similar to a typical water vapour concentration in the atmosphere. A very good agreement has been obtained between the calibration factors for CH3O2 detection with two different concentrations of water vapour in the flow tube:  $7.5 \times 10^{16}$  molecule cm-3 or  $3.0 \times 10^{17}$  molecule cm-3 (corresponding to  $2.6 \times 10^{14}$ molecule cm-3 and  $1.0 \times 10^{15}$  molecule cm-3, respectively in the FAGE cell) as shown in Figure 6 in Sect. 2.3.2.1. This very good agreement for H2O vapour and the above calculations for CH4 support the use of the flow tube method for the FAGE calibration of the CH3O2 concentrations."

**Minor comments**

P4 L4: "Here we report he first ..." should read "Here we report the first ..."

The suggested correction has been made.

**P4 L13: Please report the sampling flow rate of the FAGE apparatus**

Now the sampling flow rate is given at the beginning of section 2.1 (page 4): "The gas was sampled with a flow rate of 3.2 slm through a 1 mm diameter pinhole..."

**P5 L4-5**: Since the detection of the CH3O fluorescence is red-shifted from the excitation, why is the counting window delayed by 100 ns from the laser pulse? This time gating approach is usually used for the detection of on-resonant fluorescence.

The off-resonance CH3O fluorescence occurs between ~ 300 - 400 nm and, hence a relatively broad bandpass filter, with an average transmission > 80% between 320–430 nm, was used for the fluorescence collection. However, it appears that red-shifted scattered laser light (the excitation wavelength was ~ 298 nm) produced in the FAGE chamber also passed through the interference filter, increasing the background. In order to avoid the majority of these background counts, the gate unit was opened 100 ns after the probe light pulse. As the optimum gate-width found for the CH3O fluorescence was 2 µs (*vide infra*), no significant loss of CH3O signal was encountered by the 100 ns delay in the fluorescence detection. Future improvements to the instrument will improve changing the cell material or coating to reduce this scattered light background.

The second paragraph on page 5 was changed as follows:

"The relatively broad bandpass filter used for the collection of the CH3O fluorescence (average transmission > 80% between 320–430 nm) allowed some red-shifted scattered light (presumably from the walls of the chamber) generated by the probe laser to be transmitted and hence detected by the MCP-PMT. In order to ameliorate this and reduce the background signal, the gate unit was opened 100 ns after the laser pulse to detect fluorescence integrated over a gate-width of 2  $\mu$ s. The optimum gate-width of 2  $\mu$ s (values in the range 1-3  $\mu$ s were compared) is consistent with the CH3O fluorescence lifetimes, calculated to be in the range of 0.9 – 1.5  $\mu$ s, using the reported radiative lifetimes for CH3O of 1.5  $\mu$ s (Inoue et al., 1979), 2.2  $\mu$ s (Ebata et al., 1982) and (4 ± 2)  $\mu$ s (Wendt and Hunziker, 1979) and using the fluorescence quenching rate coefficients of N2 and O2 (Wantuck et al., 1987) to calculate the rate of quenching at the pressure in the FAGE detection cell ((2.65 ± 0.05) Torr). As the fluorescence lifetime of CH3O(A) in the detection cell is 0.9–1.5  $\mu$ s, delaying the counting of the fluorescence by 100 ns makes very little difference (~ 10%) in the fraction of fluorescence collected."

**P5** L12-15: The authors mention that the wavelength is tuned on/off resonance with the CH3O transition line. In FAGE instruments, OH is continuously generated in a reference cell to be able to precisely tune the laser wavelength on and off resonance. How is it performed for CH3O on this instrument? Is CH3O continuously generated in a reference cell? If so, how is it done?

The signals were large enough that during conditions where  $CH_3O_2$  concentrations were constant (e.g. in calibrations or during HIRAC experiments where steady-state concentrations were generated) it was established that the laser-wavelength was stable over a long period once the laser wavelength had been tuned to the CH3O transition. Hence, the online wavelength position for CH3O fluorescence detection was found without using a reference cell. The laser excitation scans shown in Figures 2 and 3 were performed using the flow tube method described in the sections 2.3.1 and 2.3.2.1 to generate either CH3O (by the CH3OH photolysis at 185 nm) or CH3O2 (by the H2O photolysis at 185 nm to generate OH followed by the reaction of the produced OH with CH4 in the presence of O2).

In the HIRAC experiments the concentration of  $CH_3O_2$  radicals generated in the chamber in a steady-state with the UV lamps turned on at the beginning of each experiment using the  $Cl_2/CH_4/air$  system was used to tune the laser at the correct excitation wavelength by performing similar scans to the laser scans shown in Figure 3.

In all measurements the offline wavelength position was fixed to the value obtained by adding 2.5 nm to  $\lambda$ (online) as described in the third paragraph on page 5. For field measurements in the future, when the concentrations of CH3O2 (and hence CH3O after conversion) will be both lower and more variable over short timescales, a reference cell will be necessary. We are in the process of developing a reference cell.

The third paragraph on page 5 was changed to clarify how the laser is tuned to the correct CH3O excitation wavelength:

"...Figure 2 shows the laser excitation spectrum centred at ~298 nm in the  $v_3$  vibronic band recorded using an increment of  $\Delta \lambda = 10^{-3}$  nm. The spectrum agrees well with previous work (Inoue et al., 1980;Kappert and Temps, 1989;Shannon et al., 2013). Figure 3 shows typical laser excitation scans performed over a narrower range of wavelengths in order to locate  $\lambda$ (online). The LIF spectra were obtained by using the CH3O or CH3O2 radicals generated in a flow tube described in Sect. 2.3.1, with the flow tube output impinged close to the FAGE sampling inlet. The radicals were generated using the 184.9 nm light output of a Hg Pen-Ray lamp by either the photolysis of methanol in nitrogen to generate CH3O or the photolysis of water vapour in synthetic air (to generate OH) in the presence of methane to form CH3O2. The CH3O radicals were directly detected, while the CH3O2 radicals were first converted to CH3O species by added NO prior to the fluorescence detection cell (Fig. 1). Similar laser scans to the scans shown in Fig. 3 were recorded by using the CH3O2 radicals produced in a steady-state concentration in HIRAC using photolytic mixtures of Cl2/CH4/air as described in Sect. 2.3.2.2. There were no unexpected features in the laser excitation scans for CH3O recorded when FAGE sampled CH3O2 radicals from HIRAC, consistent with no interference being anticipated in the FAGE measurements of CH3O as there were no other species in HIRAC absorbing at 298 nm and fluorescing at the wavelengths transmitted by the bandpass filter (average transmission > 80 % over 320 - 430 nm).

In this work the FAGE signals were large enough that during conditions where  $CH_3O_2$  concentrations were constant (e.g. in calibrations or during HIRAC experiments where steady-state concentrations were generated) it was established that the laser wavelength was stable over a long period once  $\lambda$  had been tuned to the CH3O transition. Hence,  $\lambda$ (online) was found without using a reference cell. We are in the process of developing a reference cell for field measurements in the future, when the concentrations of CH3O2 (and hence CH3O after conversion) will be both lower and more variable over short timescales."

In addition, the future construction of the reference cell is mentioned in the paragraph of section 3.1.1 where all the future instrument improvements are listed:

"The further optimizations of sensitivity and the planned construction of a reference cell to find the online wavelength position could potentially enable  $CH_3O_2$  measurements to be made in urban environments where  $CH_3O_2$  concentrations are estimated to be considerably lower, for example a few 107 molecule cm-3 based on modeling results (Whalley et al., to be submitted)."

**P7** L15: The authors indicate a CH3O2-to-CH3O conversion efficiency of 40% at the optimum NO concentration. However, since CH3O can also be lost through its reaction with NO (and potentially through its reaction with O2 as well), isn't the 40% representative of a lower limit of the conversion?

The 40% value represents the optimum  $CH_3O_2$  to  $CH_3O$  conversion efficiency as  $CH_3O$  is rapidly formed (by the  $CH_3O_2$  + NO reaction) and removed in the system (by the  $CH_3O$  reactions with NO and  $O_2$ ), as discussed in section 2.2 (page 7). The text in section 2.2 explains that this result was obtained by comparison of the FAGE signal vs. [NO] generated by numerical simulations using a chemistry system formed by the above reactions with experimental data. Therefore, no text change has been made as the value of 40% was obtained from a simulation at the relevant conditions.

**P10 L12-13**: It is indicated that the photon flux was varied between 0-1.5E14 photon/cm2/s. However, the lower bound reported for the radical generation is 1.5E10 molecule/cm3, which cannot correspond to a photon flux set at zero. Please clarify.

The lower limit of the photon flux was corrected:

"The concentration of CH3O2 was varied by changing the photon flux in the range of  $0.5-1.5 \times 10^{14}$  photon cm-2 s-1 to generate [CH3O2] =  $1.5-4.5 \times 10^{10}$  molecule cm-3."

**P12 L9-14**: The detection limits are calculated for a BKG signal of approximately 100 ct/s, which is reported as a typical value for this instrument. What are the contributions of the scattered visible and laser lights? How is the BKG signal expected to change when the solar irradiation changes during field measurements? How will it affect the detection limit during daytime?

The contributions to be background are roughly 50% laser scattered light within the detection cell and 50% visible light which enters the pinhole. For field measurements, there will be a contribution from solar scattered light which will scale with sunlight intensity. As for measurements of OH, the detection limit depends on the standard deviation of the background signal, and for more intense solar radiation, this will increase, increasing the detection limit. The visible scattered light is recorded on its own (together with dark counts) in a separate photon collection integration gate which is delayed a long time after the laser pulse, and is subtracted from the counts from the integration gate containing the fluorescence (after scaling for any differences in the two gate widths).

The second paragraph on page 12 was modified:

"...BKG is the background signal and had a typical value of ~100 counts s-1, which represents ~50 counts s-1 laser scattered light within the detection cell and ~50 counts s-1 scattered visible light which enters the pinhole from the room with a negligible contribution (1 count s-1 on average) of the detector dark counts, *t* is the time per data point, *m* represents the number of online data points and *n* is the number of offline data points." "cm-1" was changed into "cm-3"

**P15 L21 & L22**: Two different uncertainties are given for the on-line signal: 12% and 6%. Which one is correct?

The paragraph on page 15 discusses the different components of the total uncertainty: 12% uncertainty represents the  $2\sigma$  error in the fluorescence signal due to the uncertainty in the online wavelength position, while the 6% uncertainty is the  $2\sigma$  error of the laser power measured with the power meter. A minor change was made in the last sentence of section 3.2.1:

"...of 12 % in the online FAGE signal and 6 % uncertainty in the laser power measured by the laser power meter and used to normalize the data. The uncertainty associated with the online signal, 12 % at  $2\sigma$  level, was calculated as the average deviation of the signal value due to the error limits of  $\pm 5 \times 10^{-4}$  nm in the online wavelength position (see the typical laser excitation scans shown in Fig. 3)."

**P16 L39**: The authors indicate that the oxygen concentration was lowered in some experiments performed on the HIRAC chamber. Could the lower oxygen concentration lead to a different sensitivity towards CH3O due to changes in quenching rates of excited CH3O?

Line 39 of page 16 describes the methoxy radical measurement in HIRAC (section 3.3) which is shown in Figure 8. The concentration of CH3O in these HIRAC experiments was obtained by using the calibration factor for methoxy radicals, which in turn was determined using the photolysis of methanol in N2 method described in section 2.3.1. In the HIRAC experiment shown in Figure 8 O2 was only present in trace amounts  $([O_2]_{HIRAC} = 5.4 \times 10^{15} \text{ molecule cm}^{-3}$ , which corresponds to 1.8 x 1013 molecule cm-3 O2 in the fluorescence detection cell) as described in section 3.3. This [O2] is too small to produce a faster quenching rate of the CH3O LIF signal in the chamber experiment compared to the quenching rate when using pure N2, as estimated using the quenching rate coefficient of O2 reported by Wantuck et al. (1986),  $2.5 \times 10^{-11} \text{ cm}^3$  molecule-1 s-1. The following sentence was added in the first paragraph of section 3.3 for clarification:

"The concentration of CH3O during the experiment was computed by using the FAGE calibration factor for methoxy radicals generated from the photolysis of methanol in N2,  $C_{CH3O} = (5.1 \pm 2.2) \times 10^{-10}$  counts cm3 molecule-1 s-1 mW-1 (Sect. 3.1.1). The temporal profile of the CH3O is shown in Fig. 8..."

**Author response to anonymous referee #3 on "A new method for atmospheric detection of the CH3O2 radical" by L. Onel et al.**

Note: The changes in the manuscript addressing the comments of the referee #3 are highlighted in yellow below. The authors refer to the line numbers in the manuscript before revision mentioned in the comments.

The authors would like to thank anonymous referee #3 for their valuable comments to this manuscript.

**Page 4 line 30 and Page 6 Figures 2 and 3**: As with detection of OH by the LIF FAGE technique, the authors must tune the laser on and off of the CH3O transition to determine the net signal due to CH3O fluorescence and the background signal due to laser scatter and other broadband fluorescence. OH LIF-FAGE instruments use a reference cell that generates high concentration of OH radicals to ensure that the laser is tuned to the correct frequency. It is unclear how the authors know that the laser is tuned to the correct CH3O excitation wavelength. Do they use a spectrometer to measure the wavelength, or do they have a reference cell that generates?

The signals were large enough that during conditions where  $CH_3O_2$  concentrations were constant (e.g. in calibrations or during HIRAC experiments where steady-state concentrations were generated) it was established that the laser-wavelength was stable over a long period once the laser wavelength had been tuned to the CH3O transition. Hence, the online wavelength position for CH3O fluorescence detection was found without using a reference cell. The laser excitation scans shown in Figures 2 and 3 were performed using the flow tube method described in the sections 2.3.1 and 2.3.2.1 to generate either CH3O (by the CH3OH photolysis at 185 nm) or CH3O2 (by the H2O photolysis at 185 nm to generate OH followed by the reaction of the produced OH with CH4 in the presence of O2).

In the HIRAC experiments the concentration of  $CH_3O_2$  radicals generated in the chamber in a steady-state with the UV lamps turned on at the beginning of each experiment using the  $Cl_2/CH_4/air$  system was used to

tune the laser at the correct excitation wavelength by performing similar scans to the laser scans shown in Figure 3.

In all measurements the offline wavelength position was fixed to the value obtained by adding 2.5 nm to  $\lambda$ (online) as described at page 5, line 14. For field measurements in the future, when the concentrations of CH3O2 (and hence CH3O after conversion) will be both lower and more variable over short timescales, a reference cell will be necessary. We are in the process of developing a reference cell.

**The third paragraph on page 5 (lines 10-24) was changed for clarification:**

"The signals were large enough that during conditions where CH3O2 concentrations were constant (e.g. in calibrations or during HIRAC experiments where steady-state concentrations were generated) it was established that the laser-wavelength was stable over a long period once the laser wavelength had been tuned to the CH3O transition. Hence, the online wavelength position for CH3O fluorescence detection was found without using a reference cell. Figure 2 shows the laser excitation spectrum centred at ~298 nm in the  $v_3$ vibronic band recorded using an increment of  $\Delta \lambda = 10^{-3}$  nm. The spectrum agrees well with previous work (Inoue et al., 1980; Kappert and Temps, 1989; Shannon et al., 2013). Figure 3 shows typical laser excitation scans performed over a narrower range of wavelengths in order to locate  $\lambda$ (online). The LIF spectra were obtained by using the  $CH_3O$  or  $CH_3O_2$  radicals generated in a flow tube described in Sect. 2.3.1, with the flow tube output impinged close to the FAGE sampling inlet. The radicals were generated using the 184.9 nm light output of a Hg Pen-Ray lamp by either the photolysis of methanol in nitrogen to generate CH3O or the photolysis of water vapour in synthetic air (to generate OH) in the presence of methane to form CH3O2. The CH3O radicals were directly detected, while the CH3O2 radicals were first converted to CH3O species by added NO prior to the fluorescence detection cell (Fig. 1). Similar laser scans to the scans shown in Fig. 3 were recorded by using the CH3O2 radicals produced in a steady-state concentration in HIRAC using photolytic mixtures of Cl2/CH4/air as described in Sect. 2.3.2.2. There were no unexpected features in the laser scans recorded when FAGE sampled CH3O2 radicals from HIRAC, consistent with no interference being anticipated in the FAGE measurements of CH3O as there were no other species in HIRAC absorbing at 298 nm and fluorescing at the wavelengths transmitted by the bandpass filter (average transmission > 80 % over 320 - 430 nm).

In this work the FAGE signals were large enough that during conditions where  $CH_3O_2$  concentrations were constant (e.g. in calibrations or during HIRAC experiments where steady-state concentrations were generated) it was established that the laser wavelength was stable over a long period once  $\lambda$  had been tuned to the  $CH_3O$  transition. Hence,  $\lambda$ (online) was found without using a reference cell. We are in the process of developing a reference cell for field measurements in the future, when the concentrations of  $CH_3O_2$  (and hence  $CH_3O$  after conversion) will be both lower and more variable over short timescales."

**Page 8 line 25**: Equation 2 assumes that the concentration of methanol is proportional to the concentration of water vapor and that any loss of methanol in their bubbler system is equal to any loss of water in their flow tube. Can the authors justify this assumption?

Equation 2 assumes that the concentration of methanol vapour in the photolysis flow tube is equal to the concentration of water vapour in the flow tube obtained when the bubbler contained water instead of methanol. The flow tube calibration using the water vapour photolysis represents the conventional FAGE calibration method for OH and  $HO_2$  and previous investigations have shown that the water vapour loss in the system formed by the bubbler and the flow tube is negligible. Even less wall losses can be expected in the case of methanol, which has a significantly higher vapour pressure than water.

The following sentence was added after equation 2 (page 8, line 29):

"Equation 2 assumes that there were no losses of water vapour and methanol vapour by condensation in the tubing connecting the bubbler to the flow tube. This is as expected based on the small difference in temperature between the bubbler (*vide supra*) and the connecting tubing (typically held at ~ 20 °C) and as the gas going through the bubbler was diluted with the gas by-passing the bubbler."

*Page 12, line 25*: (*i*) *The authors claim that reducing the pressure in their FAGE detection cell could increase the sensitivity of the instrument. Is this due to reduced quenching of the CH3O fluorescence by air? (ii) Have the authors measured the impact trace gases on the fluorescence efficiency, such as water vapor?*

(i) A potential improvement of the instrument sensitivity for  $CH_3O_2$  by using a pressure in the detection cell lower than the present limit of 2.65 Torr is expected because of the experimental observation of an increase in the fluorescence signal when the pressure in the detection cell is reduced from 10.00–2.65 Torr. As the pressure is reduced there is a reduction in the CH3O number density (which would decrease the LIF signal) and also a decrease in the quenching rate of the CH3O fluorescence by air, and hence an increase in the fluorescence quantum yield (which would increase the LIF signal). These two effects are opposing, but at low pressures do not cancel, leading to the observed increase in signal with lower pressures. It is therefore expected that as the pressure is reduced further below 2.65 Torr that the signal would continue to increase. Another reason could be that the characteristics of the jet expansion and/or the ensuing flow to the LIF detection region change with pressure, leading to a more favourable transmission of radicals to the detection region, but it is difficult to test this experimentally. For clarification the text (page 12, lines 24 - 27) was modified as follows: "The present investigations into the change of sensitivity with pressure in the range from 2.65-10.00 Torr found that 2.65 Torr is the optimum value in this pressure interval. The result suggests that, by reducing the pressure in the above range of values, the decrease in fluorescence due to the reduction in the CH3O number density was overcome by the increase in the fluorescence quantum yield due to a lower fluorescence quenching rate. Another reason could be that the characteristics of the jet expansion and/or the ensuing flow to the LIF detection region change with pressure, leading to a more favourable transmission of radicals to the detection region, but it is difficult to test this experimentally. Hence an additional improvement in the sensitivity might be obtained by using a lower detection cell pressure than the current value of 2.65 Torr using a more powerful pump."

(ii) No measurement of the rate coefficients of the fluorescence quenching by the traces gases has been performed in this work. However, a very good agreement was obtained between the flow tube calibrations for  $CH_3O_2$  with two different concentrations of water vapour in the flow tube: 7.5 x  $10^{16}$  molecule cm-3 or 3 x  $10^{17}$  molecule cm-3 (corresponding to 2.6 x  $10^{14}$  molecule cm-3 and 1.0 x  $10^{15}$  molecule cm-3, respectively in the FAGE detection cell) as shown by Figure 6 in Sect. 2.3.2.1. The result presented in Figure 6 shows that the CH3O fluorescence quenching rate by water is minor for the above [H2O].

Methane was also present in the FAGE chamber in concentrations of several times  $10^{14}$  molecule cm-3. Calculations using the CH3O fluorescence quenching rate coefficient of CH4 reported by Wantuck et al. (1987),  $1.05 \times 10^{-10}$  s-1, and a pressure in the FAGE detection cell of 2.65 Torr show only minor decreases in the fluorescence quantum yield, by few percent, when [CH4] is increased from zero to the experimental values. Assuming a quenching rate coefficient of H2O equal to that of CH4, similar small decreases in the fluorescence quantum yield when [H2O] was increased from zero to the concentration values used in the flow tube calibration (0.3 - 1.0 x  $10^{15}$  molecule cm-3).

A paragraph which discusses the  $CH_3O(A)$  quenching rates of water and methane at the concentrations used in the flow tube calibration of the FAGE instrument for  $CH_3O_2$  has been added at the end of the section 3.1.1:

"The calibrations using the flow-tube ("wand") method have been performed under water vapour concentrations similar to the ambient  $[H_2O_{vapour}]$  but few orders of magnitude higher than those present in the HIRAC chamber experiments. In contrast with  $[H_2O_{vapour}]$  the methane concentrations used in the "wand" method were similar to [CH4] present in HIRAC but higher than [CH4] in the atmosphere. However, as detailed in this paragraph, the effects of methane and water on our sensitivity are minimal. Estimations using the reported fluorescence quenching rate coefficient of CH3O(*A*) by CH4,  $k_{quench.CH4} = 1.05 \times 10^{-10} \text{ s}^{-1}$ , (Wantuck et al., 1987) and the concentrations of CH4 in the LIF detection cell for the calibrations using the flow-tube  $(1.7 \times 10^{14} \text{ molecule cm}^{-3} \text{ and } 3.4 \times 10^{14} \text{ molecule cm}^{-3}$ , corresponding to  $5.0 \times 10^{16} \text{ molecule cm}^{-3}$  and  $1.0 \times 10^{17} \text{ molecule cm}^{-3}$ , respectively in the flow tube) resulted in only ~ 1–2% lower fluorescence

quantum yield compared to the value determined in the absence of CH4. No literature value has been found for the fluorescence rate coefficient of CH3O(*A*) fluorescence by H2O vapour. However, even if it assumed to be as large as the above reported value for CH4 ( $k_{quench.CH4}$ ), only a few percent decrease in the fluorescence quantum yield is computed (compared with a water concentration of zero) for the levels of H2O vapour which are present at the CH3O2 FAGE detection axis when using the flow tube calibration method. These levels (1– 2 % v/v) are similar to a typical water vapour concentration in the atmosphere. A very good agreement has been obtained between the calibration factors for CH3O2 detection with two different concentrations of water vapour in the flow tube:  $7.5 \times 10^{16}$  molecule cm-3 or  $3.0 \times 10^{17}$  molecule cm-3 (corresponding to  $2.6 \times 10^{14}$ molecule cm-3 and  $1.0 \times 10^{15}$  molecule cm-3, respectively in the FAGE cell) as shown in Figure 6 in Sect. 2.3.2.1. This very good agreement for H2O vapour and the above calculations for CH4 support the use of the flow tube method for the FAGE calibration of the CH3O2 concentrations."

**Page 12, line 26**: How does the OH sensitivity of the HIRAC FAGE compare to the field instrument? Assuming the CH3O sensitivity scales with the differences in the OH sensitivity, can the authors be more specific regarding the potential improvement in the LOD if this technique were to be used in the field instrument?

The HIRAC FAGE sensitivity for OH is about two times lower than the ground-based field instrument sensitivity for OH:  $C_{OH (HIRAC)} = 8 \times 10^{-8}$  counts cm3 molecule-1 s-1 mW-1,  $C_{OH (field)} = 1.5 \times 10^{-7}$  counts cm3 molecule-1 s-1 mW-1.

As the distance from the inlet pinhole to the laser axis in the  $CH_3O_2$  fluorescence cell (Figure 1, 580 mm) is considerably longer than the corresponding distance in the ground–based field fluorescence cell for OH and HO2 detection (88 mm), improvements in the  $CH_3O_2$  sensitivity are expected for the field FAGE instrument. The decrease in the pinhole–to–laser axis from 580 mm to 88 mm would result in a reduced loss of the  $CH_3O_2$  radicals on the instrument internal walls and would provide a greater population in the laser probed rotational level as the gas is still cooler than ambient following the pinhole expansion. However, the increase in the  $CH_3O$  sensitivity cannot be quantified simply using the difference in the OH sensitivity between the HIRAC instrument and the field instrument. A larger increase in sensitivity between the field instrument and HIRAC would be expected for OH than for  $CH_3O_2$  based on expected heterogeneous losses, as the wall loss of OH is larger than the wall loss of  $CH_3O_2$ . However, how much the decrease in temperature at the laser axis owing to a smaller nozzle-to-laser axis distance improves the FAGE sensitivity for  $CH_3O$  compared to the sensitivity for OH needs further investigation.

We think that no modification of the text is necessary as it cannot be assumed that improvements in the  $CH_3O_2$  sensitivity will scale with the difference in the OH sensitivity between the HIRAC instrument and the ground-field instrument.

**Page 16, line 19**: The authors suggest that based on their flow tube calibrations that the rate constant for the CH3O2 + CH3O2 reaction may be 25% too high, perhaps due to a 25% overestimation of the CH3O2 absorption cross section. What is the uncertainty associated with the recommended rate constant? Does the rate constant derived using their flow tube calibration factor agree to within the combined uncertainty of the calibration and the rate constant?

The associated uncertainty with the IUPAC recommended value of the rate coefficient for the CH3O2 self-reaction,  $k_{CH3O2}$ , is ~ 12% (1 $\sigma$ ). Our measured value, based on the flow tube calibration factor is ~25% lower than the IUPAC recommendation, with an overall error of ~20% (1 $\sigma$ ). Therefore, the obtained  $k_{CH3O2}$  have overlapping error limits with the IUPAC preferred value at the 1 $\sigma$  level.

The overall uncertainties of the two calibration methods of FAGE are discussed in detail in the manuscript. Even though the kinetic method agrees well with the flow tube method, it should be noted that the use of a lower value of *k* than  $k_{CH3O2}$ (IUPAC) would improve the level of agreement. Therefore, the text has not been changed.

**Page 17, Figure 8**: The authors measure the concentration of CH3O in nitrogen to reduce the loss of CH3O from the CH3O2 + O2 reaction. However, it appears that they use the calibration factor determined in air to estimate the CH3O concentrations in this experiment. Does the calibration factor change in N2 compared to air due to different fluorescence quenching rates?

The concentration of CH3O in the HIRAC experiment shown in Figure 8 was obtained by using the calibration factor for methoxy radicals, which in turn was determined using the photolysis of methanol in N2 method described in section 2.3.1. In this HIRAC experiment O2 was only present in trace amounts ( $[O_2]_{HIRAC} = 5.4 \times 10^{15}$  molecule cm-3, which corresponds to 1.8 x 1013 molecule cm-3 O2 in the fluorescence detection cell) as described in section 3.3. This  $[O_2]$  is too small to produce a faster quenching rate of the CH3O LIF signal in the chamber experiment compared to the quenching rate when using pure N2, as estimated using the quenching rate coefficient of O2 reported by Wantuck et al. (1986),  $2.5 \times 10^{-11}$  cm3 molecule-1 s-1.

The following sentence was added in the first paragraph of section 3.3 for clarification:

"The concentration of CH3O during the experiment was computed by using the FAGE calibration factor for methoxy radicals generated from the photolysis of methanol in N2,  $C_{CH3O} = (5.1 \pm 2.2) \times 10^{-10}$ counts cm3 molecule-1 s-1 mW-1 (Sect. 3.1.1). The temporal profile of the CH3O is shown in Fig. 8..."

**A new method for atmospheric detection of the CH3O2 radical**

Lavinia Onel1, Alexander Brennan1, Paul W. Seakins1,2, Lisa Whalley1,2, Dwayne E. Heard1,2

1School of Chemistry, University of Leeds, Leeds, LS2 9JT, UK

2National Centre for Atmospheric Science, University of Leeds, LS2 9JT, UK

Correspondence to: Lavinia Onel (chmlo@leeds.ac.uk); Dwayne Heard (d.e.heard@leeds.ac.uk)

Abstract. A new method for measurement of the methyl peroxy (CH3O2) radical has been developed using the conversion of CH3O2 into CH3O by excess NO with subsequent detection of CH3O by fluorescence assay by gas expansion (FAGE) with laser excitation at *ca.* 298 nm. The method can also directly detect CH3O, when no nitric oxide is added. Laboratory calibrations were performed to characterise the FAGE instrument sensitivity using the conventional radical source employed in OH calibration with conversion of a known concentration of OH into CH3O2 via reaction with CH4/O2. Detection limits of  $3.8 \times 10^8$  molecule cm-3 and  $3.0 \times 10^8$  molecule cm-3 were determined for CH3O2 and CH3O2 to  $1.1 \times 10^8$  molecule cm-3 comparable to atmospheric concentrations. The kinetics of the second–order decay of CH3O2 via its self–reaction were observed in HIRAC (Highly Instrumented Reactor for Atmospheric Chemistry) at 295 K and 1 bar and used as an alternative method of calibration. The overall uncertainties of the two methods of calibrations are similar: 15 % for the kinetic method and 17 % for the conventional method and are discussed in detail. The capability to quantitatively measure CH3O in chamber experiments is demonstrated via observation in HIRAC of CH3O formed as a product of the CH3O2 self–reaction.

**1** Introduction**

Methyl peroxy ( $CH_3O_2$ ) radicals are critical intermediates in the atmospheric oxidation (Orlando and Tyndall, 2012) and combustion of hydrocarbons (Zador et al., 2011). In the remote atmosphere  $CH_3O_2$  is mainly formed by the reaction of methane with the OH radical via abstraction of an H atom (R1), followed by the reaction of the produced  $CH_3$  radical with  $O_2$  (R2).

$$OH + CH_4 \rightarrow CH_3 + H_2O$$

$$CH_3 + O_2 + M \rightarrow CH_3O_2 + M$$
(R1)
(R2)

Methyl radicals can also be formed from more complex species, e.g. the reaction of acetyl peroxy radicals with HO2 in low NOx environments or the reaction of acetyl peroxy radicals with NO in anthropogenically influenced environments. CH3O2 is predicted to be the most abundant peroxy radical in the atmosphere, yet there are no specific measurements of its concentration. Daytime concentrations estimated using a box model utilizing the MCM (Master Chemical Mechanism) version 3.3.1 (Saunders et al., 2003;Jenkin et al., 2015) are ~ 6 × 108 molecule cm-3 in the tropical Atlantic ocean in summer (Whalley et al., 2010), ~ 2 × 108 molecule cm-3 in a tropical rainforest (Whalley et al., 2011), and lower in polluted environments, for example ~ 5 × 107 molecule cm-3 in London in summertime (Whalley et al., to be submitted).

The reaction of  $CH_3O_2$  with NO (R3) usually dominates the chemistry of  $CH_3O_2$ , particularly in environments influenced by anthropogenic NOx emissions, resulting in NO2 production and hence ozone production:

$$CH_3O_2 + NO \rightarrow CH_3O + NO_2$$
 (R3)

The subsequent reaction of  $CH_3O$  with  $O_2$  (R4) produces  $HO_2$ , which in turn oxidises another NO to  $NO_2$  (R5) with further production of  $O_3$  and propagation of the  $HO_x$  radical chain:

$$CH_{3}O + O_{2} \rightarrow CH_{2}O + HO_{2}$$

$$HO_{2} + NO \rightarrow OH + NO_{2}$$
(R4)
(R5)

However, under low NOx levels (e.g. remote forested environments and the marine boundary layer) the self-reaction of  $CH_3O_2$  (R6) and the reactions of  $CH_3O_2$  with HO2 and other organic peroxy (RO2) species are important radical removal/termination reactions. The  $CH_3O_2$  self-reaction occurs through two channels, (R6.a) and (R6.b) (Tyndall et al., 1998):

$$CH_{3}O_{2} + CH_{3}O_{2} \rightarrow CH_{3}OH + CH_{2}O + O_{2}$$

$$CH_{3}O_{2} + CH_{3}O_{2} \rightarrow CH_{3}O + CH_{3}O + O_{2}$$

$$(R6.b)$$

Despite the importance of the reaction (R6), there are uncertainties of about a factor of two in the value of its rate coefficient at room temperature,  $k_6$ , which ranges from (2.7–5.2) × 10-13 cm3 molecule-1 s-1 (Atkinson et al., 2006); the preferred IUPAC value is  $k_6 = 3.5 \times 10^{-13}$  cm3 molecule-1 s-1 (Atkinson et al., 2006). The previous kinetic studies used time-resolved UV-absorption spectroscopy to detect CH3O2 radical, typically at 250 nm, (Sander and Watson, 1980, 1981;McAdam et al., 1987;Kurylo and Wallington, 1987;Jenkin et al., 1988;Simon et al., 1990;Lightfoot et al., 1990). UV-absorption spectroscopy is a relatively insensitive technique and hence the detection limits of CH3O2 were quite high, for example approximately  $4 \times 10^{12}$  molecule cm-3 (Sander and Watson, 1980, 1981). In addition, due to the broad, featureless spectra of RO2 species, which often overlap, UV-absorption is a relatively unselective technique for the study of the kinetics of individual RO2. Therefore, there is a clear need for the determination of  $k_6$  using a more selective method, which will be addressed in subsequent studies.

At present, CH3O2 is not specifically measured in the atmosphere by any direct or indirect method. Time-resolved continuouswave cavity ringdown spectroscopy (CRDS), using the  $v_{12}$  transition of the  $A \leftarrow X$  band at ~ 1.3 µm has been used to detect CH3O2 directly in a photoreactor (Farago et al., 2013;Bossolasco et al., 2014). However, the detection limit is not sufficiently sensitive to enable tropospheric detection. Typically, the sum of  $HO_2$  and all organic  $RO_2$  has been measured in the atmosphere, making no distinction between HO2 and different RO2 species, although more recently the sum of RO2 has been quantified separately to HO2. One of the methods uses Chemical Ionisation Mass Spectrometry to determine the sum  $[HO_2] + \sum_i [RO_{2,i}]$  or separately  $[HO_2]$ , depending on the control of the flows of the NO and SO2 reagents (Hanke et al., 2002;Edwards et al., 2003). The sum  $[HO_2]$  +  $\sum_{i} [RO_{2,i}]$  has also been determined for many years by the Peroxy Radical Chemical Amplifier (PERCA) method, which uses NO and CO to generate NO2 amplified by a chain reaction, and subsequently measured by a variety of methods, for example luminol fluorescence, laser-induced fluorescence (LIF) or cavity absorption methods (Cantrell and Stedman, 1982;Cantrell et al., 1984; Miyazaki et al., 2010; Hernandez et al., 2001; Green et al., 2006; Chen et al., 2016). A modification of PERCA, using a denuder to remove HO2 has been used to estimate the sum of RO2 (Miyazaki et al., 2010). ROxLIF is a more recent method, which uses OH LIF detection at low pressure, known as FAGE (fluorescence assay by gas expansion) (Fuchs et al., 2008; Whalley et al., 2013). The ROxLIF method measures either  $[HO_x] = [OH] + [HO_2]$  by converting HOx into HO2 through addition of CO, or  $[RO_x] =$  $[HO_x] + \sum_i ([RO_{2,i}] + [RO_i])$  by titrating ROx to HO2 by added NO and CO. After the conversion into HO2, HO2 is converted into OH in the FAGE chamber and detected by LIF. The sum  $\sum_{i} [RO_{2,i}]$  and the concentration of the initial HO2 can be determined from the separate measurements of HOx, ROx and OH. The limit of detection of the ROxLIF method is ~ 0.1 pptv ( $2.5 \times 10^6$  molecule cm- 3) (Fuchs et al., 2008; Whalley et al., 2013). Recently, the interference from certain types of RO2 radicals in the FAGE detection of HO2 was deliberately exploited to enable a partial RO2 
[revised manuscript text omitted]
 2  $\mu$ s. The optimum gate-width of 2  $\mu$ s (values in the range 1-3  $\mu$ s were compared) is consistent with the CH3O fluorescence lifetimes, calculated to be in the range of 0.9 – 1.5  $\mu$ s, using the reported radiative lifetimes for CH3O of 1.5  $\mu$ s (Inoue et al., 1979), 2.2  $\mu$ s (Ebata et al., 1982) and (4 ± 2)  $\mu$ s (Wendt and Hunziker, 1979) and using the fluorescence quenching rate coefficients of N2 and O2 (Wantuck et al., 1987) to calculate the rate of quenching at the pressure in the FAGE detection cell ((2.65 ± 0.05) Torr). As the fluorescence lifetime of CH3O(A) in the detection cell was 0.9–1.5  $\mu$ s, delaying the counting of the fluorescence by 100 ns makes very little difference (~ 10%) in the fraction of fluorescence collected.

All LIF signals reported here were normalized to the probe laser power as measured with a laser power meter (Maestro, Gentec-EO) before the start of each LIF measurement. Fluctuations in the relative laser power were monitored via a photodiode (UDT-555UV, Laser Components) during the measurements and were accounted for in the signal normalization. The LIF spectrum was corrected for the laser-scattered background by subtracting the normalized offline signal recorded over 60 s at the end of each LIF measurement using an offline wavelength  $\lambda$ (offline = 300.29 nm) =  $\lambda$ (online = 297.79 nm) + 2.5 nm, well away from any CH3O absorption. The signals were large enough that during conditions where  $CH_3O_2$  concentrations were constant (e.g. in calibrations or during HIRAC experiments where steady-state concentrations were generated) it was established that the laser-wavelength was stable over a long period once the laser wavelength had been tuned to the CH3O transition. Hence, the online wavelength position for CH3O fluorescence detection was found without using a reference cell. Figure 2 shows the laser excitation spectrum centred at ~298 nm in the  $v_3$  vibronic band recorded using an increment of  $\Delta \lambda = 10^{-3}$  nm. The spectrum agrees well with previous work (Inoue et al., 1980;Kappert and Temps, 1989;Shannon et al., 2013). Figure 3 shows typical laser excitation scans performed over a narrower range of wavelengths in order to locate  $\lambda$ (online). The LIF spectra were obtained by using the CH3O or CH3O2 radicals generated in a flow tube described in Sect. 2.3.1, with the flow tube output impinged close to the FAGE sampling inlet. The radicals were generated using the 184.9 nm light output of a Hg Pen-Ray lamp by either the photolysis of methanol in nitrogen to generate  $CH_{3}O$ or the photolysis of water vapour in synthetic air (to generate OH) in the presence of methane to form  $CH_3O_2$ . The  $CH_3O$  radicals were directly detected, while the CH3O2 radicals were first converted to CH3O species by added NO prior to the fluorescence detection cell (Fig. 1). Similar laser scans to the scans shown in Fig. 3 were recorded by using the CH3O2 radicals produced in a steady-state concentration in HIRAC using photolytic mixtures of Cl2/CH4/air as described in Sect. 2.3.2.2. There were no unexpected features in the laser scans recorded when FAGE sampled CH3O2 radicals from HIRAC, consistent with no interference being anticipated in the FAGE measurements of CH3O as there were no other species in HIRAC absorbing at 298 nm and fluorescing at the wavelengths transmitted by the bandpass filter (average transmission > 80 % over 320 - 430 nm).

In this work the FAGE signals were large enough that during conditions where  $CH_3O_2$  concentrations were constant (e.g. in calibrations or during HIRAC experiments where steady-state concentrations were generated) it was established that the laser wavelength was stable over a long period once  $\lambda$  had been tuned to the CH3O transition. Hence,  $\lambda$ (online) was found without using a reference cell. We are in the process of developing a reference cell for field measurements in the future, when the concentrations of CH3O2 (and hence CH3O after conversion) will be both lower and more variable over short timescales.